# CLUSTERFORMER: Clustering As A Universal Visual Learner

**James C. Liang**
Rochester Institute of Technology

**Yiming Cui**
University of Florida

**Qifan Wang**
Meta AI

**Tong Geng**
University of Rochester

**Wenguan Wang**
Zhejiang University

**Dongfang Liu**[*]
Rochester Institute of Technology

## Abstract

This paper presents CLUSTERFORMER, a universal vision model that is based on the CLUSTERing paradigm with TransFORMER. It comprises two novel designs: ① *recurrent cross-attention clustering*, which reformulates the cross-attention mechanism in Transformer and enables recursive updates of cluster centers to facilitate strong representation learning; and ② *feature dispatching*, which uses the updated cluster centers to redistribute image features through similarity-based metrics, resulting in a transparent pipeline. This elegant design streamlines an explainable and transferable workflow, capable of tackling heterogeneous vision tasks (*i.e.*, image classification, object detection, and image segmentation) with varying levels of clustering granularity (*i.e.*, image-, box-, and pixel-level). Empirical results demonstrate that CLUSTERFORMER outperforms various well-known specialized architectures, achieving 83.41% top-1 acc. over ImageNet-1K for image classification, 54.2% and 47.0% mAP over MS COCO for object detection and instance segmentation, 52.4% mIoU over ADE20K for semantic segmentation, and 55.8% PQ over COCO Panoptic for panoptic segmentation. For its efficacy, we hope our work can catalyze a paradigm shift in universal models in computer vision.

## 1 Introduction

Computer vision has seen the emergence of specialized solutions for different vision tasks (*e.g.*, ResNet [34] for image classification, Faster RCNN [70] for object detection, and Mask RCNN [33] for instance segmenta-

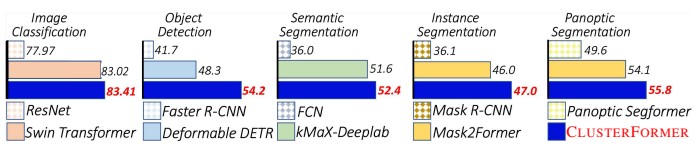

Figure 1: CLUSTERFORMER is a clustering-based universal model, offering superior performance over various specialized architectures.

tion), aiming for superior performance. Nonetheless, neuroscience research [73, 65, 82, 5] has shown that the human perceptual system exhibits exceptional interpretive capabilities for complex visual stimuli, without task-specific constraints. This trait of human perceptual cognition diverges from current computer vision techniques [95, 44, 46], which often employ diverse architectural designs.

Human vision possesses a unique attention mechanism that selectively focuses on relevant parts of the visual field while disregarding irrelevant information [81, 40]. This can be likened to a clustering approach [2, 3, 89], in which individual pixel points are decomposed and reorganized into relevant concepts to address various tasks. This essentially is a hierarchical process that involves combining

---

[*]Corresponding author.

37th Conference on Neural Information Processing Systems (NeurIPS 2023).

basic visual features, such as lines, shapes, and colors, to create higher-level abstractions of objects, scenes, and individuals [79, 59, 66, 27] . Inspired by the remarkable abilities of the human vision system, this work aims to develop a universal vision model that can replicate the unparalleled prowess.

To this end, we employ a clustering-based strategy that operates at varying levels of granularity for visual comprehension. By solving different vision tasks (*i.e.*, image classification, object detection, and image segmentation), we take into account the specificity at which visual information is grouped (*i.e.*, image-, box-, and pixel-level). We name our approach, CLUSTERFORMER (§3.2), as it utilizes a CLUSTERing mechanism integrated within the TransFORMER architecture to create a universal network. The method begins by embedding images into discrete tokens, representing essential features that are grouped into distinct clusters. The cluster centers are then recursively updated through a recurrent clustering cross-attention mechanism that considers associated feature representations along the center dimension. Once center assignments and updates are complete, features are dispatched based on updated cluster centers, and then both are fed into the task head for the target tasks.

CLUSTERFORMER enjoys a few attractive qualities. ❶ *Flexibility*: CLUSTERFORMER is a clustering-anchored approach that accommodates a broad array of visual tasks with superior performance (see Fig. 1) under one umbrella. The core epistemology is to handle various tasks with different levels of granularity (*e.g.*, image-level classification, box-level detection, pixel-level segmentation, *etc.*), moving towards a universal visual solution. ❷ *Transferability*: The cluster centers generated by the CLUSTERFORMER encoder are directly employed by the task head as initial queries for clustering, allowing the entire architecture to transfer underlying representation for target-task predictions (see Table. 4). This elegant design facilitates the transferability of knowledge acquired from the upstream task (*i.e.*, encoder trained on ImageNet [72]) to downstream tasks (*e.g.*, decoder trained on instance segmentation on COCO [49]). ❸ *Explainability*: Regardless of the target tasks, CLUSTERFORMER's decision-making process is characterized by a transparent pipeline that continuously updates cluster centers through similarity-based metrics. Since the reasoning process is naturally derivable, the model inference behavior is ad-hoc explainable (see §4.2). This differs CLUSTERFORMER from most existing unified models [17, 44, 95] that fail to elucidate precisely how a model works.

To effectively assess our method, we experimentally show: In §4.1.1, with the task of image classification, CLUSTERFORMER outperforms traditional counterparts, *e.g.*, $0.13 \sim 0.39\%$ top-1 accuracy compared with Swin Transformoer [53] on ImageNet [72], by training from scratch. In §4.1.2, when using our ImageNet-pretrained, our method can be expanded to the task of object detection and greatly improve the performance compared to Dino [96] over Swin Transformer on COCO [49] ($0.8 \sim 1.1\%$ mAP). In addition, our method can also adapt to more generic per-pixel tasks, *a.k.a*, semantic segmentation (see §4.1.3), instance segmentation (see §4.1.4), and panoptic segmentation (see §4.1.5). For instance, we achieve performance gains of $0.6 \sim 1.3\%$ mIoU for semantic segmentation on ADE20K [101], $1.0 \sim 1.4\%$ mAP for instance segmentation on MS COCO [49] and $1.5 \sim 1.7\%$ PQ for panoptic segmentation on COCO Panoptic [42] compared with Mask2Former [17] over Swin Transformer. Our algorithm are extensively tested, and the efficacy for the core components is also demonstrated through a series of ablative studies outlined in §4.2,

## 2   Related Work

**Universal Vision Model.** Transformers [81] have been instrumental in driving universal ambition, fostering models that are capable of tackling tasks of different specificity with the same architecture and embody the potential of these recent developments [23, 17, 16, 95, 96, 4, 80, 30, 57, 86] in the field. In the vision regime, mainstream research endeavors have been concentrating on the development of either encoders [53, 88] or decoders [44, 94]. The encoder is centered around the effort of developing foundation models [4, 53, 24, 22], trained on extensive data that can be adapted and fine-tuned to diverse downstream tasks. For instance, Swin Transformer [53] capably serves as a general-purpose backbone for computer vision by employing a hierarchical structure consisting of shifted windows; ViT-22B [22], parameterizes the architecture to 22 billion and achieves superior performance on a variety of vision tasks through learning large-scale data. Conversely, research on decoders [23, 17, 16, 95, 94, 44, 96, 87, 50, 20, 52, 19, 21, 51, 93, 76, 37, 99, 25, 48] is designed to tackle homogeneous target tasks, by using queries to depict visual patterns. For instance, Mask2Former [17] incorporates mask information into the Transformer architecture and unifies various segmentation tasks (*e.g.*, semantic, instance, and panoptic segmentation); Mask-DINO [44] extends the decoding process from detection to segmentation by directly utilizing query embeddings

for target task predictions. Conceptually different, we streamline an elegant systemic workflow based on clustering and handle heterogeneous visual tasks (*e.g.*, image classification, object detection, and image segmentation) at different clustering granularities.

**Clustering in Vision.** Traditional clustering algorithms in vision [39, 28, 29, 55, 91, 1, 10, 61, 6, 58] can be categorized into the hierarchical and partitional modes. The hierarchical methods [62, 38] involve the modeling of pixel hierarchy and the iterative partitioning and merging of pixel pairs into clusters until reaching a state of saturation. This approach obviates the necessity of a priori determination of cluster quantity and circumvents the predicaments arising from local optima. [98, 12]. However, it exclusively considers the adjacent pixels at each stage and lacks the capacity to assimilate prior information regarding the global configuration or dimensions of the clusters. [69, 64]. In contrast, partitional clustering algorithms [78, 36] directly generate a flat structure with a predetermined number of clusters and exclusively assign pixels to a single cluster. This design exhibits a dynamic nature, allowing pixels to transition between clusters [11, 63]. By employing suitable measures, this approach can effectively integrate complex knowledge within cluster centers. As a powerful system, human vision incorporates the advantages of both clustering modes [89, 83, 67]. We possess the capability of grouping analogous entities at different scales. Meanwhile, we can also effectively categorize objects purely based on their shape, color, or texture, without having the hierarchical information. Drawing on the above insights, we reformulate the attention mechanism (§3.2 ) in Transformer architectures [81] from the clustering's perspective to decipher the hierarchy of visual complexity.

# 3 Methodology

## 3.1 Preliminary

**Clustering.** The objective of clustering is to partition a set of data points, denoted by $X \in \mathbb{R}^{n \times d}$, into $C$ distinct clusters based on their intrinsic similarities while ensuring that each data point belongs to only one cluster. Achieving this requires optimizing the stratification of the data points, taking into account both their feature and positional information, to form coherent and meaningful groupings. Clustering methodologies typically employ advanced similarity metrics, such as cosine similarity, to measure the proximity between data points and cluster centroids. Additionally, they consider the spatial locality of the points to make more precise group assignments.

**Cross-Attention for Generic Clustering.** Drawing inspiration from the Transformer decoder architecture [81], contemporary end-to-end architecture [17, 9] utilize a query-based approach in which a set of $K$ queries, $\boldsymbol{C} = [\boldsymbol{c}_1; \cdots; \boldsymbol{c}_K] \in \mathbb{R}^{K \times D}$, are learned and updated by a series of cross-attention blocks. In this context, we rethink the term "$\boldsymbol{C}$" to associate queries with cluster centers at each layer. Specifically, cross-attention is employed at each layer to adaptively aggregate image features and subsequently update the queries:

$$\boldsymbol{C} \leftarrow \boldsymbol{C} + \mathrm{softmax}_{HW}(\boldsymbol{Q}^C(\boldsymbol{K}^I)^\top)\boldsymbol{V}^I, \tag{1}$$

where $\boldsymbol{Q}^C \in \mathbb{R}^{K \times D}, \boldsymbol{V}^I \in \mathbb{R}^{HW \times D}, \boldsymbol{K}^I \in \mathbb{R}^{HW \times D}$ represent linearly projected features for query, key, and value, respectively. The superscripts "$C$" and "$I$" denote the features projected from the center and image features, respectively. Motivated by [95], we follow a reinterpretation of the cross-attention mechanism as a clustering solver by considering queries as cluster centers and applying the *softmax* function along the query dimension ($K$) instead of the image resolution ($HW$):

$$\boldsymbol{C} \leftarrow \boldsymbol{C} + \mathrm{softmax}_K(\boldsymbol{Q}^C(\boldsymbol{K}^I)^\top)\boldsymbol{V}^I. \tag{2}$$

## 3.2 CLUSTERFORMER

In this subsection, we present CLUSTERFORMER (see Fig. 2(a)). The model has a serial of hierarchical stages that enables multi-scale representation learning for universal adaptation. At each stage, image patches are tokenized into feature embedding [81, 53, 24], which are grouped into distinct clusters via a unified pipeline — first *recurrent cross-attention clustering* and then *feature dispatching*.

**Recurrent Cross-Attention Clustering.** Considering the feature embeddings $\boldsymbol{I} \in \mathbb{R}^{HW \times D}$ and initial centers $\boldsymbol{C}^{(0)}$, we encapsulate the iterative Expectation-Maximization (EM) clustering process,

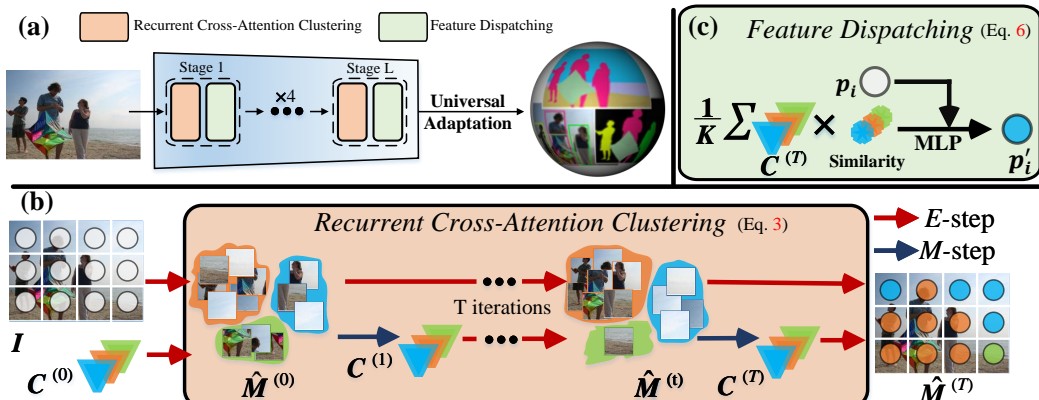

Figure 2: (a) Overall pipeline of CLUSTERFORMER. (b) Each *Recurrent Cross-Attention Clustering* layer carries out $T$ iterations of cross-attention clustering (*E*-step) and center updating (*M*-step) (see Eq. 3). (c) The *feature dispatching* redistributes the feature embeddings on the top of updated cluster centers (see Eq. 6).

consisting of $T$ iterations, within a *Recurrent EM Cross-Attention* layer (see Fig. 2(b)):

$$
\begin{aligned}
E\text{-step:} \quad & \hat{\boldsymbol{M}}^{(t)} = \mathrm{softmax}_K(\boldsymbol{Q}^{C^{(t)}}(\boldsymbol{K}^I)^\top), \\
M\text{-step:} \quad & \boldsymbol{C}^{(t+1)} = \hat{\boldsymbol{M}}^{(t)}\boldsymbol{V}^I \in \mathbb{R}^{K \times D},
\end{aligned}
\tag{3}
$$

where $t \in \{1, \cdots, T\}$ and $\hat{\boldsymbol{M}} \in [0,1]^{K \times HW}$ represents the "soft" cluster assignment matrix (*i.e.*, probability maps of $K$ clusters). As defined in Section 3.1, $\boldsymbol{Q}^C \in \mathbb{R}^{K \times D}$ denotes the query vector projected from the center $\boldsymbol{C}$, and $\boldsymbol{V}^I, \boldsymbol{K}^I \in \mathbb{R}^{HW \times D}$ correspond to the value and key vectors, respectively, projected from the image features $\boldsymbol{I}$. The Recurrent Cross-Attention approach iteratively updates cluster membership $\hat{\boldsymbol{M}}$ (*i.e.*, *E*-step) and centers $\boldsymbol{C}$ (*i.e.*, *M*-step). This dynamic updating strategy embodies the essence of partitional clustering. It enjoys a few appealing characteristics:

- *Efficiency:* While the vanilla self-attention mechanism has a time complexity of $\mathcal{O}(H^2W^2D)$, the *Recurrent Cross-Attention* approach exhibits a lower bound of $\mathcal{O}(TKHWD)$. This is primarily due to the fact that $TK \ll HW$ (*i.e.*, 4165 in Swin [53] *vs.* 1200 in ours). Specifically, considering the nature of the pyramid architecture [88, 53] during the encoding process, $TK$ can indeed be much smaller than $HW$, especially in the earlier stages. It is important to note that during each iteration, merely the $\boldsymbol{Q}$ matrix requires an update, while the $\boldsymbol{K}$ and $\boldsymbol{V}$ matrices necessitate a single computation. Consequently, the whole model enjoys systemic efficiency (see Table 6c).

- *Transparency:* The transparency hinges on the unique role that cluster centers play in our *Recurrent Cross-Attention* mechanism. The cluster centers, derived through our clustering process, act as 'prototypes' for the features they cluster. These 'prototypes' serve as a form of a representative sample for each cluster, reflecting the most salient or characteristic features of the data points within that cluster. Moreover, the *Recurrent Cross-Attention* method adheres to the widely-established EM clustering algorithm, offering a lucid and transparent framework. This cluster center assignment behaves in a human-understandable manner (see Fig. 3) during representation learning and fosters ad-hoc explainability, allowing for a more intuitive understanding of the underlying relationships.

- *Non-parametric fashion:* The *Recurrent Cross-Attention* mechanism achieves a recursive nature by sharing the projection weights for query, key, and value across iterations. This approach effectively ensures recursiveness without the introduction of additional learnable parameters (see Table 6b).

Since the overall architecture is hierarchical, *Recurrent Cross-Attention* is able to thoroughly explore the representational granularity, which mirrors the process of hierarchical clustering:

$$
\boldsymbol{C}^l = \mathrm{RCA}^l(\boldsymbol{I}^l, \boldsymbol{C}_0^l),
\tag{4}
$$

where RCA stands for the recurrent cross-attention layer. $\boldsymbol{I}^l$ is the image feature map at different layers by standard pooling operation with $H/2^l \times W/2^l$ resolution. $\boldsymbol{C}^l$ is the cluster center matrix for $l^{th}$ layer and $\boldsymbol{C}_0^l$ is the initial centers at $l^{th}$ layer. The parameters for *Recurrent Cross-Attention* at different layers, *i.e.*, $\{\mathrm{RCA}^l\}_{l=1}^L$, are not shared. In addition, we initialize the centers from image grids:

$$
[\boldsymbol{c}_1^{(0)}; \cdots; \boldsymbol{c}_K^{(0)}] = \mathrm{FFN}(\mathrm{Adptive\_Pooling}_K(\boldsymbol{I})),
\tag{5}
$$

where FFN stands for Position-wise Feedforward Network which is an integral part of the Transformer architecture. It comprises two fully connected layers along with an activation function used in the hidden layer. $\text{Adptive\_Pooling}_K(\boldsymbol{I})$ refers to select $K$ feature centers from $\boldsymbol{I}$ using adaptive sampling, which calculates an appropriate window size to achieve a desired output size adaptively, offering more flexibility and precision compared to traditional pooling methods.

**Feature Dispatching.** After the cluster assignment, the proposed method employs an adaptive process that dispatches each patch within a cluster based on similarity (see Fig. 2(c)), leading to a more coherent and representative understanding of the overall structure and context within the cluster. For every patch embedding $p_i \in I$, the updated patch embedding $p_i^{'}$ is computed as:

$$p_i^{'} = p_i + \text{MLP}(\frac{1}{K} \sum_{k=0}^{K} sim(C_k, p_i) * C_k) \tag{6}$$

This equation represents the adaptive dispatching of feature embeddings by considering the similarity between the feature embedding and the cluster centers ($C$), weighted by their respective similarities. By incorporating the intrinsic information from the cluster centers, the method refines the feature embeddings, enhancing the overall understanding of the image's underlying structure and context. All feature representations are utilized for handling the target tasks in the decoding process. In §3.3, we discuss more details about the implementation of the ending tasks.

### 3.3 Implementation Details

The implementation details and framework of CLUSTERFORMER are shown in (Fig. 2a). We followed the architecture and configuration of Swin Transformer [53]. The code will be available at here.

- *Encoder.* The encoding process is to generate presentation hierarchy, denoted as $\{\boldsymbol{I}^l\}$ with $l = \{1, 2, 3, 4\}$, for a given image $I$. The pipeline begins with the feature embedding to convert the images into separate feature tokens. Subsequently, multi-head computing [81, 53] is employed to partition the embedded features among them. Center initialization (Eq. 5) is then adopted as a starting for initializing the cluster centers, and the recurrent cross-attention clustering (Eq. 3) is utilized to recursively update these centers. Once the centers have been updated, the features are dispatched based on their association with the updated centers (Eq. 6). The further decoding process leverage both the centers and the features, which guarantees well-rounded learning.
- *Adaptation to Image Classification.* The classification head is a single-layer Multilayer Perceptron (MLP) takes the cluster centers from the encoder for predictions.
- *Adaptation to Detection and Segmentation.* Downstream task head has six Transformer decoder layers with the core design of recurrent cross-attention clustering (Eq.4). Each layer has 3 iterations.

## 4 Experiment

We evaluate our methods over five vision tasks *viz.* image classification, object detection, semantic segmentation, instance segmentation, and panoptic segmentation on four benchmarks.

**ImageNet-1K for Image Classification**. ImageNet-1K[72] includes high-resolution images spanning distinct categories (*e.g.*, animals, plants, and vehicles). Following conventional procedures, the dataset is split into 1.2M/50K/100K images for `train/validation/test` splits.
**MS COCO for Object Detection and Instance Segmentation**. COCO [49] dataset features dense annotations for 80 common objects in daily contexts. Following standard practices [49], the dataset is split into 115K/5K/20K images for `train2017/val2017/test-dev` splits.
**ADE20K for Semantic Segmentation.** ADE20K [101] dataset offers an extensive collection of images with pixel-level annotations, containing 150 diverse object categories in both indoor and outdoor scenes. The dataset comprises 20K/2K/3K images for `train/val/test` splits.
**COCO Panoptic for Panoptic Segmentation.** The COCO Panoptic dataset [42] includes 80 "thing" categories and a carefully annotated set of 53 "stuff" categories. In line with standard practices [42], the COCO Panoptic dataset is split into 115K/5K/20K images for the `train/val/test` splits as well.

The ensuing section commences by presenting the main results of each task (§4.1), succeeded by a series of ablative studies (§4.2), which aim to confirm the efficacy of each modulating design.

## 4.1 Main Results

### 4.1.1 Experiments on Image Classification

**Training.** We use `mmclassification`[2] as codebase and follow the default training settings. The default configuration for our model involves setting the number of centers to 100. To optimize the model's performance, we employ cross-entropy as the default loss function, which is widely used in classification tasks and helps in minimizing the difference between predicted probabilities and ground truth. For the training details, we run the model for 300 epochs, allowing sufficient time for the model to learn and converge. To manage the learning rate, we initialize it at 0.001 as default. The learning rate is then scheduled using a cosine annealing policy, which gradually decreases the learning rate over time. Due to limitations in our GPU capacity, we are constrained to set the total batch size at 1024. Models are trained *from scratch* on sixteen A100 GPUs.

**Results on ImageNet.** Table 1 illustrates our compelling results over different famous methods. CLUSTERFORMER exceeds the Swin Transformer [53] by **0.13**% and **0.39**% on Tiny-based and Small-based models with fewer parameters (*i.e.*, 27.85M *vs.* 28.29M and 48.71M *vs.* 49.61M), respectively. On `top-5` accuracy, our approach also outperforms the Swin-Tiny and Swin-Small with gains of **0.71**% and **0.84**%, respectively. In addition, our margins over the ResNet family [34] are **3.44**% $\sim$ **4.76**% on top-1 accuracy with on-par parameters (*i.e.*, 27.85M *vs.* 25.56M and 48.71M *vs.* 44.55M).

Table 1: **Classification `top-1` and `top-5` accuracy** on ImageNet [72] `val` (see §4.1.1 for details).

| Method | #Params | top-1 | top-5 |
|---|---|---|---|
| Context Cluster-Tiny[ICLR23][58] | 5.3M | 71.68% | 90.49% |
| DeiT-Tiny[ICML21][80] | 5.72M | 74.50% | 92.25% |
| PViG-Tiny[NeurIPS22][31] | 9.46M | 78.38% | 94.38% |
| ResNet-50[CVPR2016][34] | 25.56M | 76.55% | 93.06% |
| Swin-Tiny[ICCV2021][53] | 28.29M | 81.18% | 95.61% |
| **CLUSTERFORMER**-Tiny | 27.85M | **81.31%** | **96.32%** |
| Context Cluster-Small[ICLR23][58] | 14.0M | 77.42% | 93.69% |
| DeiT-Small[ICML21][80] | 22.05M | 80.69% | 95.06% |
| PViG-Small[NeurIPS22][31] | 29.02M | 82.00% | 95.97% |
| ResNet-101[CVPR2016][34] | 44.55M | 77.97% | 94.06% |
| Swin-Small[ICCV2021][53] | 49.61M | 83.02% | 96.29% |
| **CLUSTERFORMER**-Small | 48.71M | **83.41%** | **97.13%** |

### 4.1.2 Experiments on Object Detection

**Training.** We use `mmdetection`[3] as codebase and follow the default training settings. For a fair comparison, we follow the training protocol in [17]: 1) the number of instances centers is set to 100; 2) a linear combination of the $\mathcal{L}_1$ loss and the GIoU Loss is used as the optimization objective for bounding box regression. Their coefficients are set to 5 and 2, respectively. In addition, the final object centers are fed into a small FFN for object classification, trained with a binary cross-entropy loss. Moreover, we set the initial learning rate to $1 \times 10^{-5}$, the training epoch to 50, and the batch size to 16. We use random scale jittering with a factor in $[0.1, 2.0]$ and a crop size of $1024 \times 1024$.
**Test.** We use one input image scale with shorter side as 800.
**Metric.** We adopt AP, $AP_{50}$, $AP_{75}$, $AP_S$, $AP_M$, and $AP_L$.
**Performance Comparison.** In Table 2, we present the numerical results for CLUSTERFORMER for object detection. We observe that it surpasses all counterparts [70, 7, 56, 77, 9, 75, 60, 102, 71, 96] with remarkable gains with respect to mAP. In particular, CLUSTERFORMER-Tiny exceeds the vanilla Deformable DETR [102], Sparse-DETR [71], and DINO [96] over Swin-T [53] by **6.5**%, **3.4**%, and **0.8**% in terms of mAP, respectively. In addition, our approach also outperforms these methods over Swin-S [53], *i.e.*, *54.2*% *vs* 48.3% *vs* 49.9% *vs* 53.3% in terms of mAP, respectively. Notably, CLUSTERFORMER achieves impressive performance without relying on additional augmentation.

### 4.1.3 Experiments on Semantic Segmentation

**Training.** We use `mmsegmentation`[4] as codebase and follow the default training settings. The training process for semantic segmentation involves setting the number of cluster centers to match the number of semantic categories, which is 150 for ADE20K [101]. Following the approach employed in recent works [97, 17, 74], we adopt a combination of the standard cross-entropy loss and an auxiliary dice loss for the loss function. By default, the coefficients for the cross-entropy and dice losses are set to 5 and 1, respectively. In addition, we configure the initial learning rate to $1 \times 10^{-5}$, the number of training epochs to 50, and the batch size to 16.

---

[2]https://github.com/open-mmlab/mmclassification
[3]https://github.com/open-mmlab/mmdetection
[4]https://github.com/open-mmlab/mmsegmentation

Table 2: Quantitative results on COCO [49] `test-dev` for **object detection** (see §4.1.2 for details).

| Algorithm | Backbone | Epoch | mAP↑ | $AP_{50}$↑ | $AP_{75}$↑ | $AP_S$↑ | $AP_M$↑ | $AP_L$↑ |
|---|---|---|---|---|---|---|---|---|
| Faster R-CNN[NeurIPS15] [70] | ResNet-101 | 36 | 41.7 | 62.3 | 45.7 | 24.7 | 46.0 | 53.2 |
| Cascade R-CNN[CVPR18] [7] | ResNet-101 | 36 | 42.8 | 61.1 | 46.7 | 24.9 | 46.5 | 56.4 |
| Grid R-CNN[CVPR19] [56] | ResNet-50 | 24 | 40.4 | 58.5 | 43.6 | 22.7 | 43.9 | 53.0 |
| EfficientDet[CVPR20] [77] | Efficient-B3 | 300 | 45.4 | 63.9 | 49.3 | 27.1 | 49.5 | 61.3 |
| DETR[ECCV20] [9] | ResNet-50 | 150 | 39.9 | 60.4 | 41.7 | 17.6 | 43.4 | 59.4 |
| Sparse R-CNN[CVPR21] [75] | ResNet-101 | 36 | 46.2 | 65.1 | 50.4 | 29.5 | 49.2 | 61.7 |
| Conditional DETR[ICCV21] [60] | ResNet-50 | 50 | 41.1 | 61.9 | 43.5 | 20.4 | 44.5 | 59.9 |
| Deformable DETR [ICLR21] [102] | Swin-T | 50 | $45.5_{\pm0.26}$ | $65.2_{\pm0.20}$ | $49.8_{\pm0.21}$ | $27.0_{\pm0.26}$ | $49.1_{\pm0.24}$ | $60.7_{\pm0.29}$ |
| | Swin-S | | $48.3_{\pm0.21}$ | $68.7_{\pm0.27}$ | $52.1_{\pm0.27}$ | $30.5_{\pm0.28}$ | $51.6_{\pm0.22}$ | $64.4_{\pm0.19}$ |
| Sparse-DETR[ICLR22] [71] | Swin-T | 50 | $48.6_{\pm0.24}$ | $69.6_{\pm0.20}$ | $53.5_{\pm0.23}$ | $30.1_{\pm0.27}$ | $51.8_{\pm0.21}$ | $64.9_{\pm0.29}$ |
| | Swin-S | | $49.9_{\pm0.21}$ | $70.3_{\pm0.27}$ | $54.0_{\pm0.26}$ | $32.5_{\pm0.22}$ | $53.6_{\pm0.28}$ | $66.2_{\pm0.25}$ |
| DINO [ICLR23] [96] | Swin-T | 50 | $51.2_{\pm0.26}$ | $68.4_{\pm0.25}$ | $55.3_{\pm0.26}$ | $31.3_{\pm0.24}$ | $55.1_{\pm0.38}$ | $65.8_{\pm0.26}$ |
| | Swin-S | | $53.3_{\pm0.27}$ | $70.9_{\pm0.38}$ | $57.6_{\pm0.23}$ | $33.8_{\pm0.23}$ | $56.4_{\pm0.32}$ | $66.9_{\pm0.26}$ |
| **CLUSTERFORMER** | Ours-Tiny | 50 | $52.0_{\pm0.32}$ | $70.4_{\pm0.25}$ | $57.5_{\pm0.32}$ | $34.2_{\pm0.28}$ | $54.8_{\pm0.29}$ | $64.8_{\pm0.22}$ |
| | Ours-Small | | $\mathbf{54.2}_{\pm0.33}$ | $\mathbf{71.8}_{\pm0.16}$ | $\mathbf{59.1}_{\pm0.17}$ | $\mathbf{35.6}_{\pm0.28}$ | $\mathbf{57.2}_{\pm0.20}$ | $\mathbf{67.4}_{\pm0.18}$ |

Furthermore, we employ random scale jittering, applying a factor within the range of [0.5, 2.0], and utilize a crop size with a fixed resolution of $640 \times 640$ pixels.

**Test.** During the testing phase, we re-scale the input image with a shorter side to 640 pixels without applying any additional data augmentation at test time.

**Metric.** Mean intersection-over-union (mIoU) is used for assessing image semantic segmentation performance.

**Performance Comparison.** Table 3 shows the results on semantic segmentation. Emprially, our method compares favorably to

Table 3: Quantitative results on ADE20K [101] `val` for **semantic segmentation** (see §4.1.3 for details).

| Algorithm | Backbone | Epoch | mIoU↑ |
|---|---|---|---|
| FCN[CVPR2015][54] | ResNet-50 | 50 | 36.0 |
| DeeplabV3+[ECCV2018][15] | ResNet-50 | 50 | 42.7 |
| APCNet[CVPR2019][32] | ResNet-50 | 100 | 43.4 |
| SETR[CVPR2021][100] | ViT-L | 100 | 49.3 |
| Segmenter[ICCV2021][74] | ViT-B | 100 | 52.1 |
| Segformer[NeurIPS2021][90] | MIT-B5 | 100 | 51.4 |
| kMaX-Deeplab[ECCV2022][95] | ConvNeXt-T | 100 | $48.3_{\pm0.15}$ |
| | ConvNeXt-S | | $51.6_{\pm0.23}$ |
| Mask2Former[CVPR2022][17] | Swin-T | 100 | $48.5_{\pm0.24}$ |
| | Swin-S | | $51.1_{\pm0.21}$ |
| **CLUSTERFORMER** | Ours-Tiny | 100 | $49.1_{\pm0.19}$ |
| | Ours-Small | | $\mathbf{52.4}_{\pm0.23}$ |

recent transformer-based approaches [54, 15, 32, 100, 74, 90, 95, 17]. For instance, CLUSTER-FORMER-Tiny surpasses both recent advancements, *i.e.*, kMaX-Deeplab [95] and Mask2Former [17] with Swin-T [53] (*i.e.*, **49.1**% *vs.* 48.3% *vs.* 48.5%), respectively. Moreover, CLUSTERFORMER-Small achieves **52.4**% mIoU and outperforms all other methods in terms of mIoU, making it competitive with *state-of-the-art* methods as well.

### 4.1.4 Experiments on Instance Segmentation

**Training.** We adopt the same training strategy for instance segmentation by following §4.1.2. For instance segmentation, we change the training objective by utilizing a combination of the binary cross-entropy loss and the dice Loss for instance mask optimization.

**Test.** We use one input image scale with a shorter side of 800.

**Metric.** We adopt AP, $AP_{50}$, $AP_{75}$, $AP_S$, $AP_M$, and $AP_L$.

**Performance Comparison.** Table 4 presents the results of CLUSTERFORMER against famous instance segmentation methods [33, 8, 14, 43, 13, 26, 23, 18, 17, 44] on COCO `test-dev`. CLUSTERFORMER shows clear performance advantages over prior arts. For example, CLUSTERFORMER-Tiny outperforms the universal counterparts Mask2Former [17] by 1.4% over Swin-T [53] in terms of mAP and on par with the *state-of-the-art* method, Mask-Dino [44] with Swin-T backbone. Moreover, CLUSTERFORMER-Small surpasses all the competitors, *e.g.*, yielding significant gains of 1.0% and 0.5% mAP compared to Mask2Former and Mask-Dino over Swin-S, respectively. Without bells and whistles, our method establishes a new *state-of-the-art* on COCO instance segmentation.

### 4.1.5 Experiments on Panoptic Segmentation

**Training.** Following the convention [84, 17], we use the following objective for network learning:

$$\mathcal{L}^{\text{Panoptic}} = \lambda^{\text{th}}\mathcal{L}^{\text{th}} + \lambda^{\text{st}}\mathcal{L}^{\text{st}} + \lambda^{\text{aux}}\mathcal{L}^{\text{aux}}, \tag{7}$$

$\mathcal{L}^{\text{th}}$ and $\mathcal{L}^{\text{st}}$ represent the loss functions for things and stuff, respectively. To ensure a fair comparison, we follow [95, 85] and incorporate an auxiliary loss calculated as a weighted sum of four different loss terms, specifically, a PQ-style loss, a mask-ID cross-entropy loss, an instance discrimination

Table 4: Quantitative results on COCO [49] `test-dev` for **instance segmentation** (see §4.1.4 for details).

| Algorithm | Backbone | Epoch | mAP↑ | AP$_{50}$↑ | AP$_{75}$↑ | AP$_S$↑ | AP$_M$↑ | AP$_L$↑ |
|---|---|---|---|---|---|---|---|---|
| Mask R-CNN[ICCV2017][33] | ResNet-101 | 12 | 36.1 | 57.5 | 38.6 | 18.8 | 39.7 | 49.5 |
| Cascade MR-CNN[PAMI2019][8] | ResNet-101 | 12 | 37.3 | 58.2 | 40.1 | 19.7 | 40.6 | 51.5 |
| HTC[CVPR2019][14] | ResNet-101 | 20 | 39.6 | 61.0 | 42.8 | 21.3 | 42.9 | 55.0 |
| PointRend[CVPR2020][43] | ResNet-50 | 12 | 36.3 | 56.9 | 38.7 | 19.8 | 39.4 | 48.5 |
| BlendMask[CVPR2020][13] | ResNet-101 | 36 | 38.4 | 60.7 | 41.3 | 18.2 | 41.5 | 53.3 |
| QueryInst[ICCV2021][26] | ResNet-101 | 36 | 41.0 | 63.3 | 44.5 | 21.7 | 44.4 | 60.7 |
| SOLQ[NeurIPS2021][23] | Swin-L$^\dagger$ | 50 | 46.7 | 72.7 | 50.6 | 29.2 | 50.1 | 60.9 |
| SparseInst[CVPR2022][18] | ResNet-50 | 36 | 37.9 | 59.2 | 40.2 | 15.7 | 39.4 | 56.9 |
| Mask2Former[CVPR2022][17] | Swin-T | 50 | 44.5$_{\pm0.16}$ | 67.3$_{\pm0.15}$ | 47.7$_{\pm0.24}$ | 23.9$_{\pm0.20}$ | 48.1$_{\pm0.16}$ | 66.4$_{\pm0.15}$ |
|  | Swin-S |  | 46.0$_{\pm0.21}$ | 68.4$_{\pm0.22}$ | 49.8$_{\pm0.24}$ | 25.4$_{\pm0.19}$ | 49.7$_{\pm0.22}$ | 67.4$_{\pm0.24}$ |
| Mask-Dino[CVPR2023][44] | Swin-T | 50 | 45.8$_{\pm0.28}$ | 69.6$_{\pm0.29}$ | 50.2$_{\pm0.26}$ | 26.0$_{\pm0.28}$ | 48.7$_{\pm0.37}$ | 66.4$_{\pm0.29}$ |
|  | Swin-S |  | 46.5$_{\pm0.39}$ | 70.1$_{\pm0.34}$ | **52.2**$_{\pm0.28}$ | **27.6**$_{\pm0.34}$ | 49.9$_{\pm0.25}$ | 69.5$_{\pm0.29}$ |
| **CLUSTERFORMER** | Ours-Tiny | 50 | 45.9$_{\pm0.26}$ | 69.1$_{\pm0.21}$ | 49.5$_{\pm0.18}$ | 25.2$_{\pm0.22}$ | 50.1$_{\pm0.24}$ | 68.8$_{\pm0.24}$ |
|  | Ours-Small |  | **47.0**$_{\pm0.19}$ | **71.5**$_{\pm0.26}$ | 51.8$_{\pm0.24}$ | 27.3$_{\pm0.16}$ | **50.5**$_{\pm0.20}$ | **72.6**$_{\pm0.22}$ |

Table 5: Quantitative results on COCO Panoptic [42] `val` for **panoptic segmentation** (see §4.1.5 for details).

| Algorithm | Backbone | Epoch | PQ↑ | PQ$^{Th}$↑ | PQ$^{St}$↑ | mAP$^{Th}_{pan}$↑ | mIoU$_{pan}$↑ |
|---|---|---|---|---|---|---|---|
| Panoptic-FPN[CVPR2019][41] | ResNet-101 | 20 | 44.0 | 52.0 | 31.9 | 34.0 | 51.5 |
| UPSNet[CVPR2019][92] | ResNet-101 | 12 | 46.2 | 52.8 | 36.5 | 36.3 | 56.9 |
| Panoptic-Deeplab[CVPR2020][16] | Xception-71 | 12 | 41.2 | 44.9 | 35.7 | 31.5 | 55.4 |
| Panoptic-FCN[CVPR2021][45] | ResNet-50 | 12 | 44.3 | 50.0 | 35.6 | 35.5 | 55.0 |
| Max-Deeplab[CVPR2021][85] | Max-L | 55 | 51.1 | 57.0 | 42.2 | – | – |
| CMT-Deeplab[CVPR2022][94] | Axial-R104$^\dagger$ | 55 | 54.1 | 58.8 | 47.1 | – | – |
| Panoptic Segformer[CVPR2022][46] | ResNet-50 | 24 | 49.6$_{\pm0.20}$ | 54.4$_{\pm0.26}$ | 42.4$_{\pm0.25}$ | 39.5$_{\pm0.20}$ | 60.8$_{\pm0.21}$ |
|  | ResNet-101 |  | 50.6$_{\pm0.21}$ | 55.5$_{\pm0.24}$ | 43.2$_{\pm0.20}$ | 40.4$_{\pm0.21}$ | 62.0$_{\pm0.22}$ |
| Mask2Former[CVPR2022][17] | Swin-T | 50 | 53.2$_{\pm0.25}$ | 59.1$_{\pm0.22}$ | 43.3$_{\pm0.23}$ | 42.3$_{\pm0.27}$ | 62.9$_{\pm0.19}$ |
|  | Swin-S |  | 54.1$_{\pm0.29}$ | 60.2$_{\pm0.28}$ | 45.6$_{\pm0.18}$ | 43.1$_{\pm0.23}$ | 63.6$_{\pm0.31}$ |
| Mask Dino[CVPR2023][44] | Swin-T | 50 | 53.6$_{\pm0.29}$ | 59.5$_{\pm0.26}$ | 44.0$_{\pm0.20}$ | 44.3$_{\pm0.29}$ | 63.2$_{\pm0.27}$ |
|  | Swin-S |  | 54.9$_{\pm0.33}$ | 61.1$_{\pm0.23}$ | 46.2$_{\pm0.26}$ | **45.0**$_{\pm0.22}$ | 64.3$_{\pm0.30}$ |
| **CLUSTERFORMER** | Ours-Tiny | 50 | 54.7$_{\pm0.22}$ | 60.8$_{\pm0.31}$ | 46.1$_{\pm0.20}$ | 43.4$_{\pm0.25}$ | 64.0$_{\pm0.20}$ |
|  | Ours-Small |  | **55.8**$_{\pm0.38}$ | **61.9**$_{\pm0.39}$ | **47.2**$_{\pm0.23}$ | 44.2$_{\pm0.22}$ | **65.5**$_{\pm0.21}$ |

loss, and a semantic segmentation loss. More information about $\mathcal{L}^{aux}$ can be found in [85, 95]. The coefficients $\lambda^{th}$, $\lambda^{st}$, and $\lambda^{aux}$ are assigned the values of 5, 3, and 1, respectively. Furthermore, the final centers are input into a small feed-forward neural network (FFN) for semantic classification, which is trained using a binary cross-entropy loss. Moreover, we set the initial learning rate to $1 \times 10^{-5}$, the number of training epochs to 50, and the batch size to 16. We also employ random scale jittering with a factor range of [0.1, 2.0] and a crop size of $1024 \times 1024$.

**Test.** We use one input image scale with a shorter side of 800.

**Metric.** We employ the PQ metric [42] and report PQ$^{Th}$ and PQ$^{St}$ for the "thing" and "stuff" classes, respectively. To ensure comprehensiveness, we also include mAP$^{Th}_{pan}$, which evaluates mean average precision on "thing" classes using instance segmentation annotations, and mIoU$_{pan}$, which calculates mIoU for semantic segmentation by merging instance masks belonging to the same category, using the same model trained for the panoptic segmentation task.

**Performance Comparison.** We perform a comprehensive comparison against two divergent groups of *state-of-the-art* methods: universal approaches [46, 17, 44] and specialized panoptic methods [41, 92, 16, 45, 85, 97, 94]. As shown in Table 5, CLUSTERFORMER outperforms both types of rivals. For instance, the performance of CLUSTERFORMER-Tiny clear ahead compared to Mask2Former [17] (*i.e.*, **54.7**% PQ *vs.* 53.2% PQ) and Mask-Dino [44] (*i.e.*, **54.7**% PQ *vs.* 53.6% PQ) on the top of Swin-T [53], and CLUSTERFORMER-Small achieves promising gains of **1.7**% and **0.9**% PQ against Mask2Former and Mask-Dino over Swin-S, respectively. Moreover, in terms of mAP$^{Th}_{pan}$ and mIoU$_{pan}$, the CLUSTERFORMER also achieves outstanding performance beyond counterpart approaches.

## 4.2 Ablative Study

This section ablates CLUSTERFORMER's key components on ImageNet [72] and MS COCO [49] `validation` split. All experiments use the tiny model.

**Key Component Analysis.** We first investigate the two major elements of CLUSTERFORMER, specifically, *Recurrent Cross-Attention Clustering* for center updating and *Feature Dispatching* for feature updating. We construct a BASELINE model without any center updating and feature dispatching technique. As shown in Table 6a, BASELINE achieves 74.59% top-1 and 91.73% top-5 accuracy. Upon applying *Recurrent Cross-Attention Clustering* to the BASELINE, we observe consistent and substantial improvements for both top-1 accuracy (74.59% → **80.57**%) and top-5 accuracy (91.73% → **95.22**%). This highlights the importance of the center updating strategy

Table 6: A set of **ablative studies** on ImageNet [72] `validation` and MS COCO [49] `test-dev` split (see §4.2). The adopted designs are marked in red.

| Algorithm Component | #Params | top-1 | top-5 |
|---|---|---|---|
| BASELINE | 21.73M | 74.59 | 91.73 |
| + Recurrent Cross-Attention Clustering | 26.27M | 80.57 | 95.22 |
| + Feature Dispatching | 23.46M | 78.58 | 94.68 |
| **CLUSTERFORMER** (**both**) | 27.85M | 81.31 | 96.32 |

(a) Key Component Analysiss

| Numbers (T) | #Params | top-1 | top-5 |
|---|---|---|---|
| 1 | | 81.06 | 96.23 |
| 2 | 27.85M | 81.22 | 96.29 |
| 3 | | 81.31 | 96.32 |
| 4 | | 81.33 | 96.33 |

(b) *Number of Recursion*

| Variant Cluster Center Updating Strategy | #Params | top-1 | top-5 |
|---|---|---|---|
| Cosine Similarity | 23.88M | 78.79 | 94.36 |
| Vanilla Cross-Attention [81] | 35.48M | 79.67 | 94.95 |
| Criss Cross-Attention [35] | 34.16M | 79.91 | 95.24 |
| $K$-Means [95] | 27.71M | 80.96 | 95.57 |
| **Recurrent Cross-Attention** | 27.85M | 81.31 | 96.32 |

(c) *Recurrent Cross-Attention Clustering*

| Head Dimension | #Params | top-1 | top-5 |
|---|---|---|---|
| 16 | 17.25M | 71.69 | 90.16 |
| 24 | 22.88M | 75.37 | 92.45 |
| **32** | 27.85M | 81.31 | 96.32 |
| 40 | 32.81M | 82.21 | 97.09 |
| 48 | 38.14M | 82.40 | 97.22 |

(d) Head Dimension

| Feature Dispatching | #Params | top-1 | top-5 |
|---|---|---|---|
| None | 26.27M | 80.57 | 95.22 |
| Vanilla FC Layer | 27.14M | 80.83 | 95.47 |
| Confidence-Based [68] | 26.81M | 80.69 | 95.30 |
| FC w/ Similarity [58] | 27.46M | 80.96 | 95.84 |
| **Ours** (Eq. 6) | 27.85M | 81.31 | 96.32 |

(e) *Feature Dispatching*

| Decoder Query Initialization | mAP↑ | $AP_{50}$↑ | $AP_{75}$↑ |
|---|---|---|---|
| Free Parameters | 44.2 | 66.3 | 46.4 |
| Direct Feature Embedding [17] | 44.5 | 67.3 | 47.2 |
| Mixed Query Selection [44] | 44.9 | 67.9 | 47.8 |
| Scene-Adoptive Embedding [47] | 45.1 | 67.8 | 48.0 |
| Centers from Encoder (**Ours**) | 45.9 | 69.1 | 49.5 |

(f) *Decoder Query Initialization* for instance segmentation

and validates the effectiveness of our approach, even without explicitly performing clustering. Furthermore, after incorporating *Feature Dispatching* into the BASELINE, we achieve significant gains of **3.99**% in top-1 accuracy and **2.95**% in top-5 accuracy. Finally, by integrating both core techniques, CLUSTERFORMER delivers the best performance across both metrics. This indicates that the proposed *Recurrent Cross-Attention Clustering* and *Feature Dispatching* can work synergistically and validates the effectiveness of our comprehensive algorithmic design.

**Recurrent Cross-attention Clustering.** We next study the impact of our *Recurrent Cross-attention Clustering* (Eq.4) by contrasting it with the cosine similarity updating, basic cross-attention [81], Criss-attention [35] and $K$-Means cross-attention [95]. As illustrated in Table 6c, our *Recurrent Cross-Attention* proves to be *effective* – it outperforms the cosine similarity, vanilla, Criss and $K$-Means by **2.52**%, **1.64**%, **1.40**% and **0.15**% top-1 accuracy respectively, and *efficient* – its #Params are significantly less than the other vanilla and Criss-attention and on par with $K$-Means, in line with our analysis in §3.2. To gain further insights into recursive clustering, we examine the effect of the recursion number $T$ in Table 6b. We discover that performance progressively improves from $81.06$% to **81.31**% in top-1 accuracy when increasing $T$ from 1 to 3, but remains constant after running additional iterations. We also observe that #Params increase as $T$ increases. Consequently, we set $T = 3$ as the default to strike an optimal balance between accuracy and computation cost.

**Multi-head Dimension.** We then ablate the head embedding dimension for the attention head in Table 6d. We find that performance significantly improves from $71.69$% to **82.40**% in top-1 accuracy when increasing the dimension from 16 to 48, but #Params steadily increase as the dimension grows. For a fair comparison with Swin [53], we set the head dimension to 32 as our default.

**Feature Dispatching.** We further analyze the influence of our *Feature Dispatching*. As outlined in Table 6e, in a standard manner without any dispatching method, the model attains $80.57$% top-1 accuracy and $95.22$% top-5 accuracy. By applying a vanilla fully connected layer to update the feature, we witness a marginal increase of **0.26**% in top-1 accuracy. Moreover, using the confidence-based updating method [68] and fully connected layer with similarity, the model demonstrates a noticeable enhancement in $0.12$% and $0.39$% top-1 accuracy, respectively. Last, our method yields significant performance advancements across both metrics, *i.e.*, **81.31**% top-1 and **96.32**% top-5 accuracy.

**Decoder Query Initialization.** Last, we examine the impact of query initialization in the decoder on a downstream task (*i.e.*, instance segmentation) in Table 6f. For free parameter initialization, the base model can achieve $44.2$% in terms of mAP. By applying direct feature embedding, the method has a slight improvement of $0.3$% mAP. In addition, the model exhibits improvements in

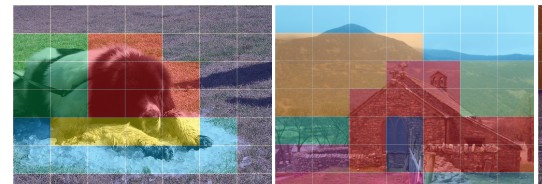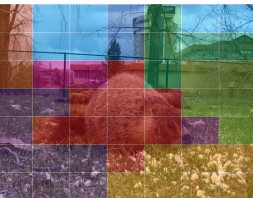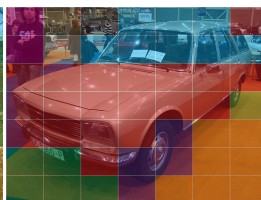

Figure 3: Visualization of center-feature assignment at the last stage of recurrent cross-attention clustering with the resolution of 7 by 7. The map displays distinct clusters, each containing features with similar representations.

mAP, achieving $44.9\%$ and $45.1\%$, respectively, by employing the mixed query selection [44] and scene-adoptive embedding [47]. Outstandingly, CLUSTERFORMER achieves the highest performance in all three metrices, *i.e.*, $45.9\%$ mAP, $69.1\%$ $AP_{50}$ and $49.5\%$ $AP_{75}$, respectively. The empirical evidence proves our design — using the cluster centers from the encoder to derive the initial query for the decoder — that facilitates the transferability for representation learning.

**Ad-hoc Explainability.** We visualize the cluster assignment map for image classification in Fig. 3. This figure provides an insightful illustration of how CLUSTERFORMER groups similar features together. Each color represents a cluster of features that share common characteristics.

## 5   Conclusion

This study adopts an epistemological perspective centered on the clustering-based paradigm, which advocates a universal vision framework named CLUSTERFORMER. This framework aims to address diverse visual tasks with varying degrees of clustering granularity. By leveraging insights from clustering, we customize the cross-attention mechanism for recursive clustering and introduce a novel method for feature dispatching. Empirical findings provide substantial evidence to support the effectiveness of this systematic approach. Based on its efficacy, we argue deductively that the proposed universal solution will have a substantial impact on the wider range of visual tasks when viewed through the lens of clustering. This question remains open for our future endeavors.

**Acknowledgement.** This research was supported by the National Science Foundation under Grant No. 2242243.

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
