# CLUSTERFORMER: Clustering As A Universal Visual Learner
## *Supplementary Material*

**James C. Liang**
Rochester Institute of Technology

**Yiming Cui**
University of Florida

**Qifan Wang**
Meta AI

**Tong Geng**
University of Rochester

**Wenguan Wang**
Zhejiang University

**Dongfang Liu**[*]
Rochester Institute of Technology

In this document, we provide additional experimental results and analysis, pseudo code, more implementation details, and discussions. It is organized as follows:

- §S1: More experimental details

- §S2: Reproducibility

- §S3: More discussions

## S1    More Experimental Details

We visualize the prediction results with the CLUSTERFORMER across a variety of datasets in §S1.1: we use the MS COCO [8] val2017 for tasks related to **object detection** and **instance segmentation**, ADE20K [14] val for **semantic segmentation**, and COCO panoptic [5] val for **panoptic segmentation**. In addition, we analyze the failure cases in §S1.2. Furthermore, we conduct additional experiments for ablative studies in §S1.3.

### S1.1    Qualitative Results

Our qualitative results (see Fig S1 for object detection, Fig S2 for semantic segmentation, Fig S3 for instance segmentation, and Fig S4 for panoptic segmentation) show that CLUSTERFORMER is capable of understanding and identifying the key traits and underlying characteristics of the images, which allows it to discern the pixel division principle from the clustering paradigm, hence yielding impressive effectiveness over various vision tasks.

### S1.2    Failure Case Analysis

The most illustrative instances of failure cases are summarized in Fig. S5. It is apparent that our algorithm finds it challenging to identify and recognize the categories of images in the following cases: ① instances with highly similar patterns, ② extremely complicated scenarios, ③ multiple objects within an image, and ④ small or severely deformed or occluded objects. To mitigate these problems, there may be a need to develop more robust clustering approaches.

---

[*]Corresponding author.

37th Conference on Neural Information Processing Systems (NeurIPS 2023).

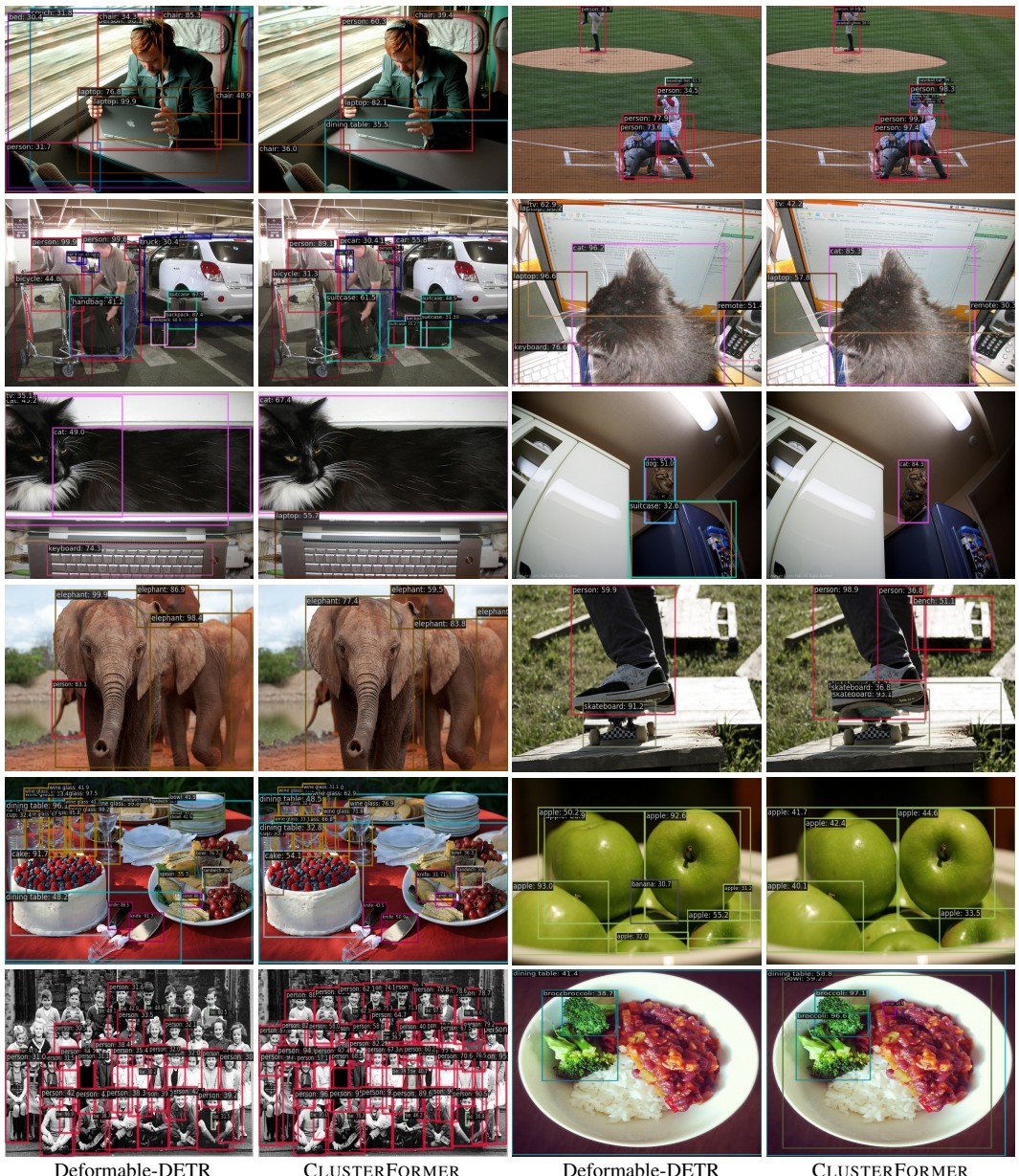

| Deformable-DETR | CLUSTERFORMER | Deformable-DETR | CLUSTERFORMER |

Figure S1: **Qualitative object detection** results on COCO [8] `val2017` compared with Deformable-DETR [15] over Swin-S [9] backbone. CLUSTERFORMER-Small achieves **54.2% *mAP***.

## S1.3   More Ablative Studies

This section provides more ablative studies of CLUSTERFORMER-Tiny on panoptic segmentation over MS COCO Panoptic [5]. In Table S1, we observe that our approach achieves consistent improvement for panoptic segmentation. The detailed explanation of each subtable can be found in §4.2.

## S2   Reproducibility

In this section, we provide pseudo-code of *Recurrent Cross-Attention Clustering* for recursive centers updating in Algorithm 1 and *Feature Dispatching* for features updating in Algorithm 2. To better ensure the reproducibility of our approach, we also include our codes at `https://anonymous.4open.science/r/mmclassification-3371/README.md` for image classification task.

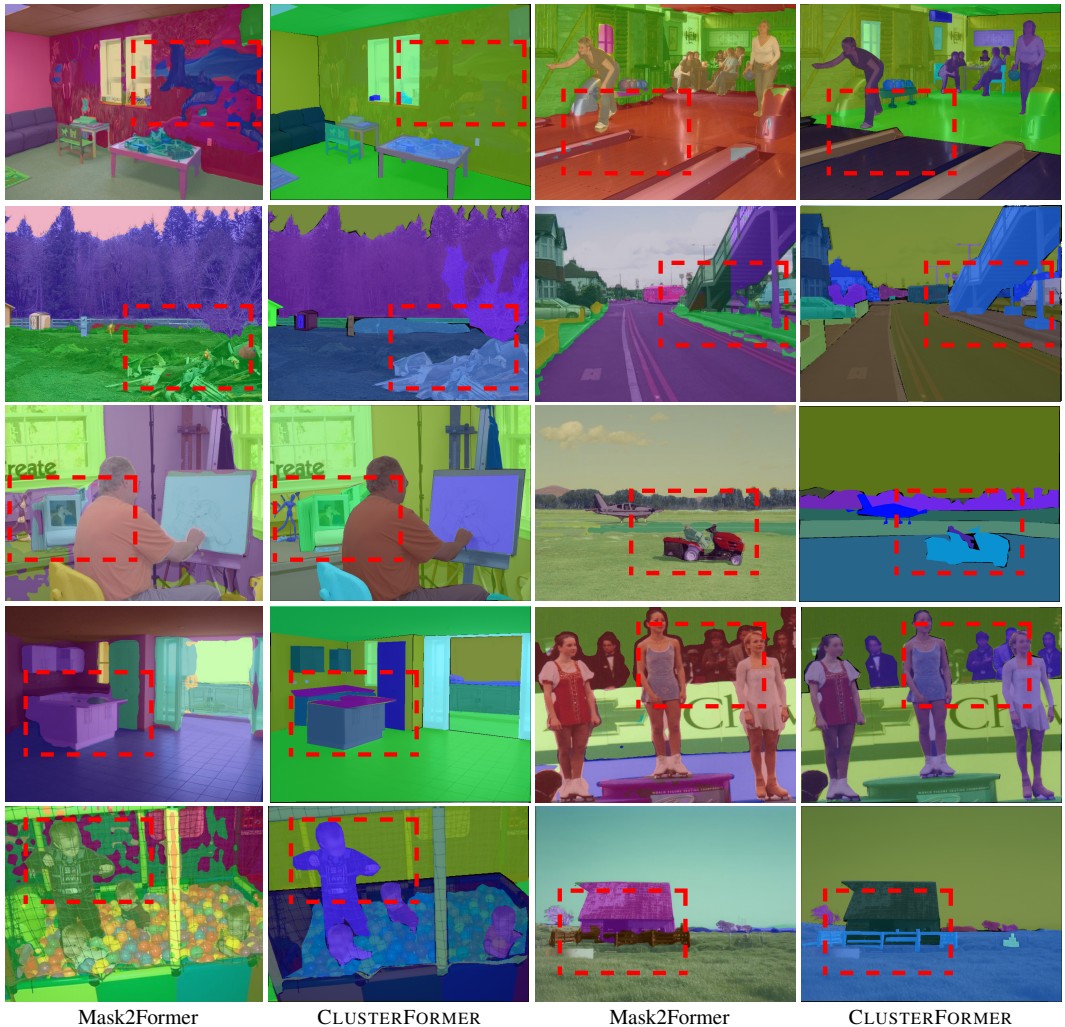

| Mask2Former | CLUSTERFORMER | Mask2Former | CLUSTERFORMER |

Figure S2: **Qualitative semantic segmentation results** on ADE20K [14] `val` compared with Mask2Former [1] over Swin-S [9] backbone. CLUSTERFORMER-Small achieves **52.4** *mIoU*.

## S3   More Discussions

### S3.1   Asset License and Consent.

We utilize three closed-set datasets across various vision tasks, *i.e.*, MS COCO [8] for object detection and instance segmentation, MS COCO Panoptic [5] for panoptic segmentation, and ADE20K [14] for semantic segmentation. They are all publicly and freely available for academic purposes. We implement all models with MMDetection [2] and MMSegmentation [3]. MS COCO (https://cocodataset.org/) is released under a CC BY 4.0; MS COCO Panoptic (https://github.com/cocodataset/panopticapi) is released under a CC BY 4.0; ADE20K (https://groups.csail.mit.edu/vision/datasets/ADE20K/) is released under a CC BSD-3; All assets mentioned above release annotations obtained from human experts with agreements. Both MMDetection (https://github.com/open-mmlab/mmdetection) and MMSegmentation (https://github.com/open-mmlab/mmsegmentation) are released under Apache-2.0.

### S3.2   Limitation

One limitation arises from the inclusion of additional clustering loops within each training iteration, which could potentially impact computation efficiency by increasing the time complexity. In our

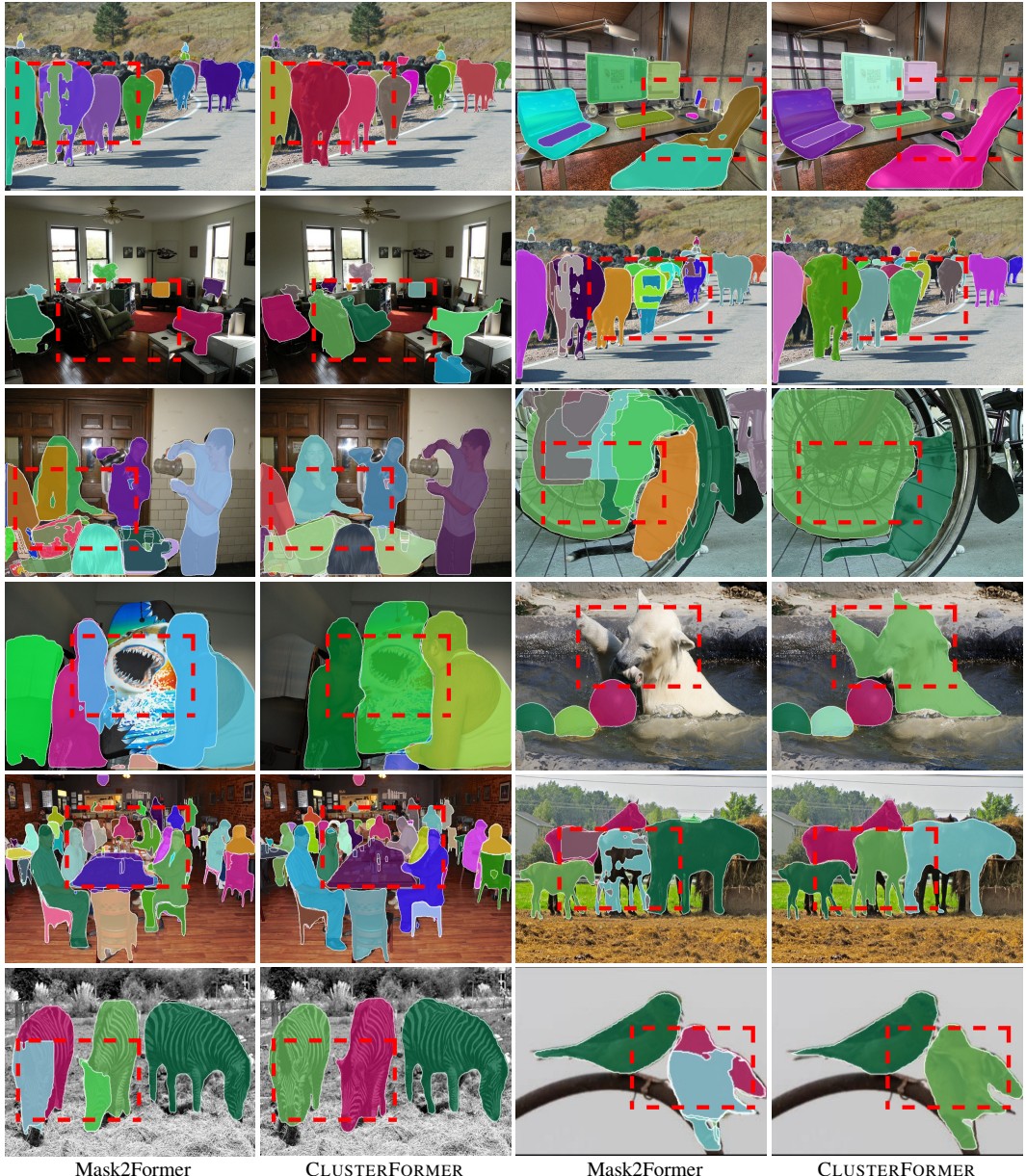

| Mask2Former | CLUSTERFORMER | Mask2Former | CLUSTERFORMER |

Figure S3: **Qualitative instance segmentation** results on COCO [8] `val2017` compared with Mask2Former [1] over Swin-S [9] backbone. CLUSTERFORMER-Small achieves **47.0%** *mAP*.

experiments, we measured a reduction of approximately 3.81% in training speed when compared with a baseline model without these recursive clustering steps. However, we observed that merely three iterations of recursive clustering are adequate to achieve global model convergence (see Table. 6b). This is a promising finding as it means that the trade-off between the depth of clustering and computation time is favorable. While this does represent an increase in computational demands, it is a relatively minor one, especially when considering the potential improvements in model performance and accuracy that our approach can deliver. We will continue to innovate and refine our methods, guided by the principles of rigorous scientific effectiveness and the pursuit of computational efficiency.

### S3.3 Broader Social Imapct

This research puts forward a universal and transparent understanding of vision tasks, connecting various tasks through a unified clustering approach. We design a unique cross-attention method for

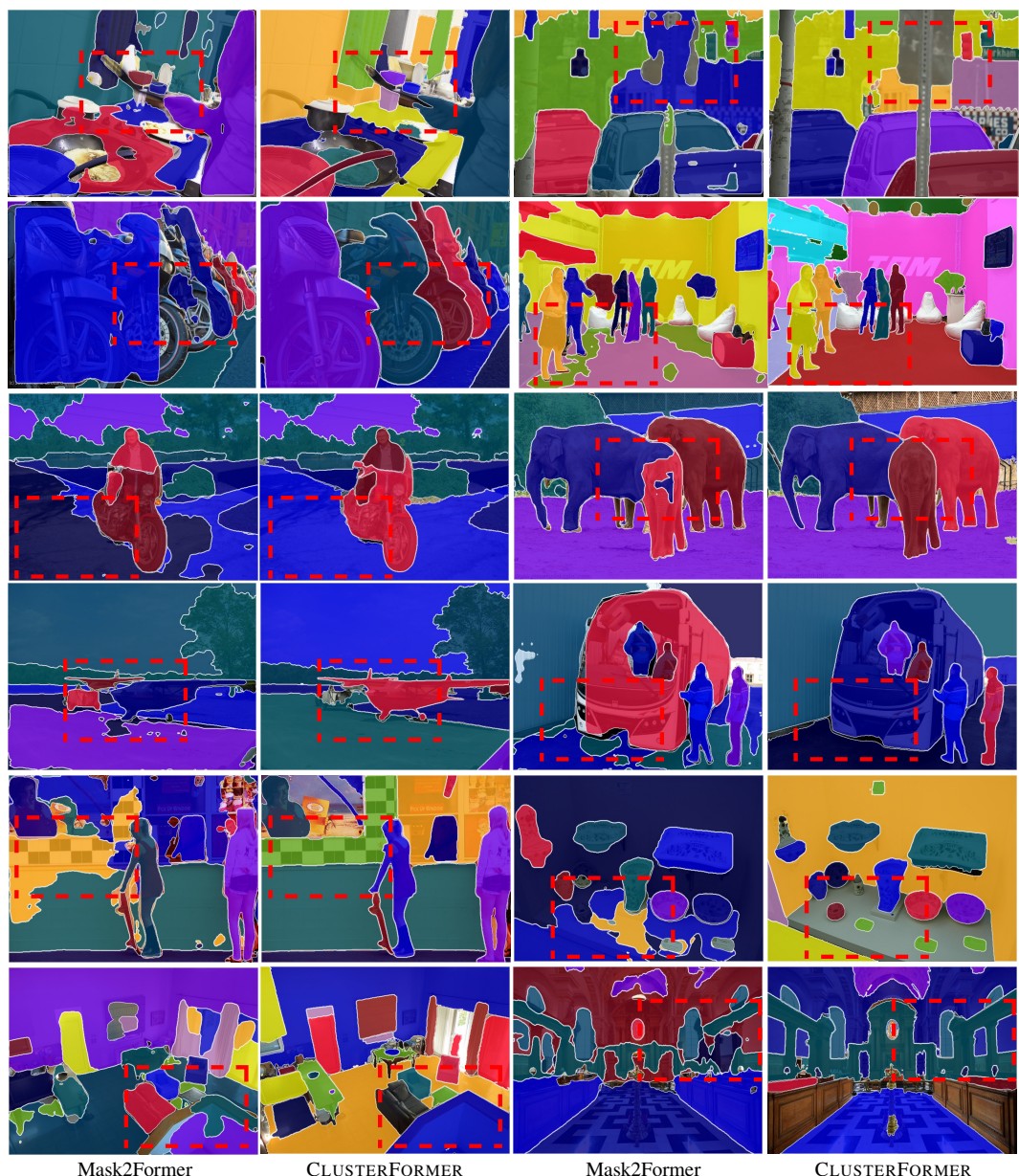

| Mask2Former | CLUSTERFORMER | Mask2Former | CLUSTERFORMER |

Figure S4: **Qualitative panoptic segmentation** results on COCO panoptic [5] val compared with Mask2Former [1] over Swin-S [9] backbone. CLUSTERFORMER-Small achieves **55.8% PQ**.

updating cluster centers, along with a neural mechanism for recurrent clustering. This optimizes the essential principles of recursive clustering for the purpose of pixel grouping. Our algorithm has shown its superiority across a range of well-known models in different vision tasks, namely object detection, instance, semantic, and panoptic segmentation. On the upside, our methodology could significantly aid a broad spectrum of real-world applications, such as autonomous driving vehicles, robotic navigation, and medical imaging. On the downside, any error or mistake in practical applications, like medical image analysis and tasks involving autonomous driving systems, may pose a risk to human safety. To mitigate this potential societal harm, we recommend the implementation of a robust safety protocol in case our methodology encounters difficulties in real-world applications.

**Similar Patterns**

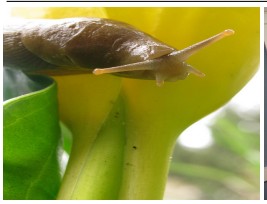 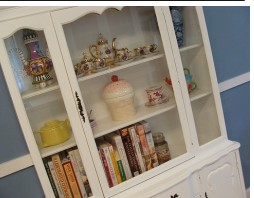

Prediction class:  snail
Ground-truth class: slug

Prediction class:  bookcase
Ground-truth class: chinese closet

**Complicated Scenarios**

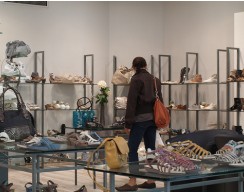 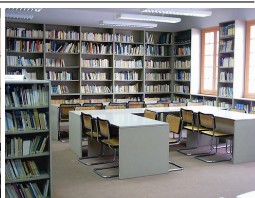

Prediction class:  restaurant
Ground-truth class: shoe store

Prediction class:  book shop
Ground-truth class: library

**Multiple Objects within an Image**

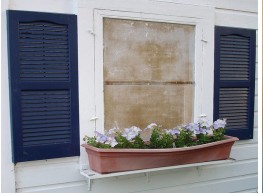 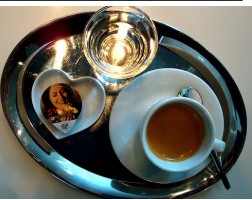

Prediction class:  rain barrel
Ground-truth class: window screen

Prediction class: ladle
Ground-truth class: tray

**Deformed or Occulded Objects**

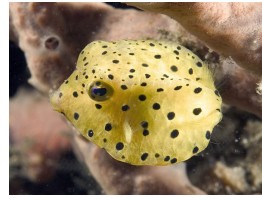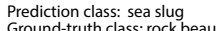 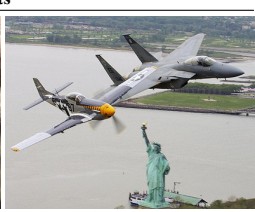

Prediction class:  sea slug
Ground-truth class: rock beauty

Prediction class:  trimaran
Ground-truth class: warplane

Figure S5: **Failure Cases** on ImageNet-1K `validation` split. See §S1.2 for details.

Table S1: A set of **ablative studies** on MS COCO Panoptic [5] `val` split (see §S1.3). The adopted designs are marked in red.

| Algorithm Component | PQ | PQ$^{Th}$ | PQ$^{St}$ |
|---|---|---|---|
| BASELINE | 49.6 | 57.5 | 38.1 |
| + Recurrent Cross-Attention Clustering | 53.6 | 59.7 | 44.5 |
| + Feature Dispatching | 51.3 | 58.2 | 42.4 |
| **CLUSTERFORMER** (**both**) | 54.7 | 60.8 | 46.1 |

(a) Key Component Analysiss

| Numbers (T) | PQ | PQ$^{Th}$ | PQ$^{St}$ |
|---|---|---|---|
| 1 | 53.3 | 59.1 | 44.6 |
| 2 | 54.2 | 60.3 | 45.5 |
| 3 | 54.7 | 60.8 | 46.1 |
| 4 | 54.9 | 70.1 | 46.2 |

(b) *Number of Recursion*

| Variant Cluster Center Updating Strategy | PQ | PQ$^{Th}$ | PQ$^{St}$ |
|---|---|---|---|
| Cosine Similarity | 51.8 | 58.4 | 42.9 |
| Vanilla Cross-Attention [12] | 53.0 | 58.9 | 44.1 |
| Criss Cross-Attention [4] | 53.3 | 59.4 | 44.3 |
| $K$-Means [13] | 54.1 | 60.2 | 45.3 |
| **Recurrent Cross-Attention** | 54.7 | 60.8 | 46.1 |

(c) *Recurrent Cross-Attention Clustering*

| Head Dimension | PQ | PQ$^{Th}$ | PQ$^{St}$ |
|---|---|---|---|
| 16 | 44.2 | 49.6 | 36.4 |
| 24 | 50.0 | 56.2 | 39.6 |
| 32 | 54.7 | 60.8 | 46.1 |
| 40 | 55.1 | 61.0 | 46.3 |
| 48 | 55.5 | 61.5 | 46.7 |

(d) Head Dimension

| Feature Dispatching | PQ | PQ$^{Th}$ | PQ$^{St}$ |
|---|---|---|---|
| None | 53.6 | 59.7 | 44.5 |
| Vanilla FC Layer | 53.8 | 60.1 | 44.9 |
| Confidence-Based [11] | 54.1 | 60.4 | 45.5 |
| FC w/ Similarity [10] | 54.2 | 60.5 | 45.7 |
| **Ours** (Eq. 6) | 54.7 | 60.8 | 46.1 |

(e) *Feature Dispatching*

| Decoder Query Initialization | PQ | PQ$^{Th}$ | PQ$^{St}$ |
|---|---|---|---|
| Free Parameters | 53.2 | 59.0 | 44.5 |
| Direct Feature Embedding [1] | 53.5 | 59.4 | 44.8 |
| Mixed Query Selection [6] | 53.9 | 59.8 | 45.1 |
| Scene-Adaptive Embedding [7] | 54.1 | 60.0 | 45.3 |
| Centers from Encoder (**Ours**) | 54.7 | 60.8 | 46.1 |

(f) *Decoder Query Initialization* for panoptic segmentation

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

**Algorithm 1** Pseudo-code of *Recurrent Cross-attention Clustering* in a PyTorch-like style.

```
"""
feats: output feature embeddings from regular projection, shape: (batch_size, channels,
    height, width)
C_0: initial cluster centers by adaptive pooling from the features, shape: (batch_size,
    num_clusters, dimension)
C: cluster centers, shape: (batch_size, num_clusters, dimension)
T: iteration number for recursive clustering
"""

# One-step cross-attention clustering in Eq.3
def recurrent_cross_attention_layer(Q, K, V):

    # E-step
    output = torch.matmul(Q, K.transpose(-2, -1))
    M = torch.nn.functional.softmax(output, dim = -2)

    # M-step
    C = torch.matmul(M, V)

    return C

# Recurrent cross-attention clustering in Eq.4
def RCross_Attention(feats, C_0, T):

    Q = nn.Linear(C_0)
    K = nn.Linear(feats)
    V = nn.Linear(feats)
    C = C_0 + recurrent_cross_attention_layer(Q, K, V)

    for _ in range(T - 1):
        Q = nn.Linear(C)
        C = C + recurrent_cross_attention_layer(Q, K, V)

    return C
```

**Algorithm 2** Pseudo-code of *Feature Dispatching* in a PyTorch-like style.

```
"""
feats: output feature embeddings from regular projection, shape: (batch_size, channels,
    height, width)
C: cluster centers, shape: (batch_size, num_clusters, dimension)
K: number of cluster centers
"""

# Feature dispatching in Eq.6
def feature_dispatching(feats, C):

    # Assign each feature to one center
    max_value, max_index = similarity(feats, C).max(dim = 1, keepdim = True)
    mask = torch.zeros_like(similarity(feats, C))
    mask.scatter_(1, max_index, 1.)
    similarity= similarity(feats, C) * mask

    # Dispatching
    feats += MLP(feats.unsqueeze(dim = 2) * similarity.unsqueeze(dim = -1)).sum(dim = 1) / K

    return feats
```

[4] Zilong Huang, Xinggang Wang, Lichao Huang, Chang Huang, Yunchao Wei, and Wenyu Liu. Ccnet: Criss-cross attention for semantic segmentation. In *ICCV*, 2019.

[5] Alexander Kirillov, Kaiming He, Ross Girshick, Carsten Rother, and Piotr Dollár. Panoptic segmentation. In *CVPR*, 2019.

[6] Feng Li, Hao Zhang, Shilong Liu, Lei Zhang, Lionel M Ni, Heung-Yeung Shum, et al. Mask dino: Towards a unified transformer-based framework for object detection and segmentation. *CVPR*, 2023.

[7] James Liang, Tianfei Zhou, and Dongfang Liu. Clustseg: Clustering for universal segmentation. In *ICML*, 2023.

[8] Tsung-Yi Lin, Michael Maire, Serge Belongie, James Hays, Pietro Perona, Deva Ramanan, Piotr Dollár, and C Lawrence Zitnick. Microsoft coco: Common objects in context. In *ECCV*, 2014.

[9] Ze Liu, Yutong Lin, Yue Cao, Han Hu, Yixuan Wei, Zheng Zhang, Stephen Lin, and Baining Guo. Swin transformer: Hierarchical vision transformer using shifted windows. In *ICCV*, 2021.

[10] Xu Ma, Yuqian Zhou, Huan Wang, Can Qin, Bin Sun, Chang Liu, and Yun Fu. Image as set of points. In *ICLR*, 2023.

[11] Yulei Qin, Juan Wen, Hao Zheng, Xiaolin Huang, Jie Yang, Ning Song, Yue-Min Zhu, Lingqian Wu, and Guang-Zhong Yang. Varifocal-net: A chromosome classification approach using deep convolutional networks. *IEEE transactions on medical imaging*, 38(11):2569–2581, 2019.

[12] Ashish Vaswani, Noam Shazeer, Niki Parmar, Jakob Uszkoreit, Llion Jones, Aidan N Gomez, Łukasz Kaiser, and Illia Polosukhin. Attention is all you need. In *NeurIPS*, 2017.

[13] Qihang Yu, Huiyu Wang, Siyuan Qiao, Maxwell Collins, Yukun Zhu, Hatwig Adam, Alan Yuille, and Liang-Chieh Chen. k-means mask transformer. *ECCV*, 2022.

[14] Bolei Zhou, Hang Zhao, Xavier Puig, Sanja Fidler, Adela Barriuso, and Antonio Torralba. Scene parsing through ade20k dataset. In *CVPR*, 2017.

[15] Xizhou Zhu, Weijie Su, Lewei Lu, Bin Li, Xiaogang Wang, and Jifeng Dai. Deformable detr: Deformable transformers for end-to-end object detection. In *ICLR*, 2021.