# OpenReview forum: "ClusterFomer: Clustering As A Universal Visual Learner"
_NeurIPS.cc/2023/Conference — NeurIPS 2023 poster_

### Official Review · Reviewer_QRN3 · 2023-06-23

**Soundness:** 3 good
**Presentation:** 3 good
**Contribution:** 2 fair
**Rating:** 6
**Confidence:** 4

**Summary:**

This paper proposes the recurrent cross-attention clustering (RCA) that groups patch features by the traditional EM algorithm with soft-assignment, and also proposes feature dispatching that aggregates spatial information using the cluster center obtained from RCA.
The novel backbone model using RCA and feature dispatching, called ClusterFormer, is proposed.
ClusterFormer shows better accuracy than baseline architectures on several datasets and tasks, including image classification, object detection, and segmentation.

**Strengths:**

- The proposed method is evaluated on major tasks in the computer vision field.
- The accuracy improvement is significant.
- It is an interesting idea to aggregate spatial information based on the proposed feature dispatching.

**Weaknesses:**

1. The technical novelty of the clustering module is limited. The hierarchical clustering with neural networks is proposed in [1] and [2], and [2] also claims the explainability. The proposed recurrent cross-attention clustering is almost the same as the superpixel sampling network[3]. The paper needs to discuss the relationship between these methods.
2. The experiments are conducted only with relatively small backbones.
3. The authors claim the explainability of the recurrent cross-attention clustering. Still, I do not understand how useful it is because I think it does not contribute to the model's explainability, i.e., we cannot know how the model makes classification results.
4. I think the paper does not provide sufficient information to implement ClusterFormer and reproduce results. At least, the detailed architectures of ClusterFormer-tiny, -small, and the overall pipeline of detection and segmentation should be described.

[1] J. Xu et al. “GroupViT: Semantic Segmentation Emerges from Text Supervision.“, 2022
[2]T. Suzuki “Clustering as Attention: Unified Image Segmentation with Hierarchical Clustering.“, 2022
[3] V. Jampani et al. “Superpixel Sampling Networks.“, 2018

**Questions:**

- Since the proposed method recurrently computes the cross-attention, I think the FLOPs and latency will be higher than the baseline methods. I would like to know the FLOPs and latency of the proposed method. Also, how much memory budget does clusterformer require?

**Limitations:**

The proposed method would require computational cost, although I do not know for sure because there is no analysis of the FLOPs and the memory budget.

---

> ### Author Rebuttal · Authors · 2023-08-09
>
> #### **Q1. The technical novelty of the clustering module**
>
> **A1:** Thank you for your suggestion, and we will discuss the relationship between these methods. Our model improves the cross-attention mechanism from an Expectation-Maximization clustering perspective to unify the encoding process. This modification provides a unique approach to the task and contributes to our model's performance. To compare with the references mentioned, [ref1] jointly trains the model and a text encoder using a paired image-text training fashion and then transfers the trained model to the task of zero-shot semantic segmentation. Our method, however, does not rely on text encoders or paired image-text training and instead focuses directly on the vision task. In [ref2], a DCN-based kernel is used to generate the attention map for clustering. While our model also uses an attention mechanism, it does so in a fundamentally different manner by modifying the cross-attention mechanism from an EM clustering perspective. Finally, [ref3] employs the superpixel sampling network, which follows the SLIC scheme. This is conceptually different from our method, where the recurrent cross-attention mechanism is used for cluster formation.
>
> We will ensure to discuss the relationship between these methods in a dedicated section of our revised manuscript, highlighting both the shared insights and distinct differences. Thank you again for the insightful suggestion.
>
> [ref1] J. Xu et al. GroupViT: Semantic Segmentation Emerges from Text Supervision, 2022
> [ref2] T. Suzuki Clustering as Attention: Unified Image Segmentation with Hierarchical Clustering, 2022
> [ref3] V. Jampani et al. Superpixel Sampling Networks, 2018
>
>
> #### **Q2. larger backbone**
>
> **A2:** Thank you for your suggestion. Due to the limited computation resources, we provide a base-sized model for image classification as follows. Based on our experience, it is generally true that increasing the size of the model -- by adding more parameters -- can lead to better performance. We conduct additional experiments on the Base-sized backbone and will incorporate the below results in the revision.
>
> |  Method  | Parameters | FLOPs | top-1 accuracy | top-5 accuracy |
> | :-: | :-: | :-: | :-: | :-: |
> | ResNet-152 |  60.19M  | 11.58G  | 78.61 | 94.15 |
> | Swin-Base |  87.77M | 15.19G  | 83.36 | 96.44 |
> | ClusterFormer-Base | 81.95M | 14.27G  | 83.62 | 97.36 |
>
> #### **Q3. Explainability**
>
> **A3:** Sorry for the confusion on the explainability. Our claim of the explainability hinges on the unique role that cluster centers play in our recurrent cross-attention mechanism. The cluster centers, derived through our clustering process, act as 'prototypes' for the features they cluster. These 'prototypes' serve as a form of a representative sample for each cluster, reflecting the most salient or characteristic features of the data points within that cluster.
>
> The benefit of this is twofold. Firstly, it provides an avenue for interpreting the clustering mechanism itself, offering a glimpse into how the data is being partitioned and what features are being considered most pertinent within each cluster. Secondly, and perhaps more significantly, these prototypes can be associated with the final classification results, providing some degree of interpretability. Specifically, by examining which cluster a particular instance is associated with, and looking at the prototype of that cluster, we can gain some insight into what features the model deemed most relevant when classifying that instance.
>
> Again, we appreciate your insight and will aim to make the level and limitations of our model's explainability more explicit in our revised version.
>
> #### **Q4. Implementation**
>
> **A4:** We appreciate your emphasis on implementation details. In response to your comment, we would like to point out that in Section 3.3 of our paper, we have endeavored to provide comprehensive information on implementation details and adaptation to different tasks. Moreover, to enhance reproducibility, we have provided both pseudo-code and the actual code via an anonymous link in our supplemental material.
>
> Concerning the specific architectures of ClusterFormer-tiny and -small, these are variations of our model followed by the same configuration (e.g., heads, embedded dimension, windows, and layers) of Swin Transformer.
>
> We appreciate your constructive feedback and will enhance the description of detailed architectures.

---

> > ### Comment · Reviewer_QRN3 · 2023-08-14
> > **Additional Questions**
> >
> > I read the authors' rebuttal, and I still have some questions.
> >
> > **1. Difference between the SLIC scheme and the recurrent cross-attention mechanism**
> >
> > I'm not sure about the difference between the SLIC scheme and the recurrent cross-attention mechanism because both are EM-based clustering.
> > In my understanding, the difference is only the design of the similarity function, and I think ClusterFormer is the architecture incorporating SSN into every downsampling layer (and I think it is an important contribution).
> > Could you clarify the conceptual difference that the authors mentioned?
> >
> > **2. Computational costs**
> >
> > I wonder why the FLOPs, GPU memory, and latency of ClusterFormer are smaller than that of Swin.
> > I thought they are larger due to the recurrent computation of the clustering.
> > If so, I think ClusterFormer with larger backbones can be trained with sixteen A100 GPUs which the authors used in the experiments.
> >
> > **3. Explainability**
> >
> > I am not sure what insight Figure 3 provides.
> > Using Fig. 3 as an example, could you explain what findings we can make?
> > And I recommend that the authors provide other examples to clarify how useful ClusterFormer is in terms of explainability.

---

> > > ### Author Response · Authors · 2023-08-16
> > > **Response to Reviewer QRN3**
> > >
> > > Thank you for the follow-up questions! We answer them as follows:
> > >
> > > #### **Q1: Difference between the SLIC scheme and the recurrent cross-attention mechanism**
> > >
> > > **A1:** Thank you for the insightful question. We try to provide some clarification (from our perspective). ClusterFormer employs the recurrent cross-attention mechanism for the purpose of clustering and center updates, while SLIC determines centers relying on distance similarity. The cross-attention methodology offers a more dynamic consideration of interactions and relationships among all features in comparison to a distance-based formulation. We're genuinely appreciative of the insightful perspective provided, and we concur with the reviewer's observation. Indeed, if we were to regard cross-attention as a form of similarity function (based on attention scores), then ClusterFormer can be viewed as an architecture that integrates SSN into each downsampling layer.
> > >
> > > #### **Q2: Computational costs**
> > >
> > > **A2:** The FLOPs of the cross-attention mechanism within a single iteration are significantly lower than in Swin Transformer. However, with more recursions, the FLOPs will increase, achieving an on-par computation cost with three iterations. To further illustrate the computational costs, we report complete results under different numbers of iterations (from 1 to 4) below.
> > >
> > > |  Number of Iterations | Parameters | FLOPs | top-1 accuracy | top-5 accuracy |
> > > | :-: | :-: | :-: | :-: | :-: |
> > > | 1 | 27.85M | 2.50G  | 81.06 | 96.23 |
> > > | 2 | 27.85M | 3.15G  | 81.22 | 96.29 |
> > > | 3 | 27.85M | 3.89G  | 81.31 | 96.32 |
> > > | 4 | 27.85M | 4.41G  | 81.33 | 96.33 |
> > >
> > > Regarding training ClusterFormer with larger backbones (e.g., ClusterFormer_large) using sixteen A100 GPUs, we genuinely appreciate your input. While this suggestion holds considerable value, due to our constrained computing resources, committing sixteen GPUs for an extensive period (over one month) for training larger backbones and fine-tuning on large datasets presents a substantial computational expense. We intend to explore more along the efficiency direction in the future to overcome this limitation.
> > >
> > > #### **Q3: Explainability**
> > >
> > > **A3:** Sorry for the confusion. Our explainability approach emphasizes ad hoc analysis. Generally speaking, dense feature vectors after a self-attention operation are highly entangled, and these vectors lack clear interpretation. In contrast, our method, viewed from a clustering perspective, directly provides features from their corresponding cluster centers, aiming to enhance systemic transparency spontaneously. This signifies our intent to provide immediate and intuitive comprehension of how the model processes classification and clusters information.
> > >
> > > In the context of Figure 3, this is evident through the cluster-anchored results. For instance, in the first image, the red cluster highlights the head of the dog, the green cluster indicates the body, and the yellow cluster pinpoints the dog's paws/legs. Such learned cluster centers allow for direct and comprehensible insight into how the model interprets and categorizes different semantics of the image.
> > >
> > > To further illustrate the explainability via visualization, we present the attention maps for several images in the classification tasks as suggested. These attention maps elucidate the correlation between the learned feature clusters and the image labels. We have shared the attention map results with the AC, as we're mindful of the constraints against uploading PDFs or providing external links during this rebuttal phase. Please don't hesitate to contact the AC for access to the results.
> > >
> > > Thank you again for the feedback, and we are glad to have further discussions.

---

> > > > ### Comment · Reviewer_QRN3 · 2023-08-16
> > > >
> > > > Thank you for your response and my concern is solved.
> > > > I hope the above discussion will be added to the revised paper.
> > > >
> > > > In my understanding, ClusterFormer has some limitations, e.g., computational costs (requiring over one month for large backbone models) and novelty (the proposed clustering module is similar to SSN, and the hierarchical clustering as downsampling is proposed in HCFormer).
> > > > However, I think the proposed method has more strengths than weaknesses.
> > > > So, I changed my rating to weak accept.

---

> > > > > ### Author Response · Authors · 2023-08-17
> > > > > **Thanks for your acknowledgment and supportive insights**
> > > > >
> > > > > Dear Reviewer,
> > > > >
> > > > > Thank you for recognizing the contribution of our approach and providing invaluable feedback. We'll ensure that all aspects of our discussion are integrated into the revised paper.
> > > > >
> > > > > Once again, we appreciate your supportive insights.
> > > > >
> > > > > Best,
> > > > >
> > > > > Authors

---

### Official Review · Reviewer_x7UH · 2023-07-04

**Soundness:** 3 good
**Presentation:** 4 excellent
**Contribution:** 3 good
**Rating:** 7
**Confidence:** 4

**Summary:**

This paper proposes a vision model based on the clustering paradigm with Transformer, named ClusterFormer.
It contains two modules, `recurrent cross-attention clustering` and `feature dispatching`.
This paper explains the `cross attention` mechanism from the perspective of `E-M` process.
It cleverly combines clustering and attention mechanisms, making feature fusion more accurate.
The `feature dispatching` module updates the patch embeddings.
The extensive experiments on classification, object detection, and image segmentation demonstrate ClusterFormer has superior accuracy.

**Strengths:**

1) Originality.
This paper explains the `cross attention` mechanism from the perspective of `E-M` process, which is novelty.
It cleverly combines clustering and attention mechanisms, making feature fusion more accurate.

2) Quality.
The vision model is carefully design and  experiments show promising performance and ablation studies present interesting results.
It presents promising results on varying levels of clustering granularity (i.e., image-, box-, and pixel-level).

3) Clarity
The paper is well-written and well-organized.
The citation of the paper is comprehensive.

4) Significance
The paper may have a high significance. It propose a new backbone paradigm, integrating clustering.
It alleviates the problem of unrelated patches being associated with each other in the global attention form of the ViT series models, and may become a universal visual learner.


**Weaknesses:**

Because of the recurrent step, the model may have a high FLOPs and slow inference speed under the same parameter size.
This paper does not reveal the inference speed, nor does it compare accuracy under the same inference speed conditions.
This may result in unfair experimental comparisons.

**Questions:**

1. The reviewer wants to see the speed of this model on different task,  e.g. using throughput (image / s).
2. The reviewer wants to know if further optimization can be made in the  initialization of the cluster center ?
3. What is the value of `k`? Is `k` related to specific task?
If `k` is different in different layers, would the results be better?
4. Can a larger model achieve better results at the same inference time?

**Limitations:**

The speed of this model may be slow.
Reviewers may question the practicality of the proposed model.

---

> ### Author Rebuttal · Authors · 2023-08-09
>
> #### **Q1. Computation cost**
>
> **A1:** The computation cost and inference speed are reported as follows.
>
> |  Method  | Parameters | FLOPs  | inference latency | GPU memory | top-1 accuracy|
> | :-: | :-: | :-: | :-: | :-: | :-: |
> | DeiT-Tiny |  5.72 M | 1.26 G  | 0.35 ms | 1884 MB | 74.50 |
> | ResNet-50 |  25.56 M | 4.12 G  | 0.96 ms | 7658MB | 76.55 |
> | Swin-Tiny |  28.29 M | 4.36 G  | 1.35 ms | 7990 MB | 81.18 |
> | ClusterFormer-Tiny | 27.85 M | 4.19 G  | 1.31 ms | 7786 MB | 81.31 |
>
> |  Method  | Parameters | FLOPs  | inference latency | GPU memory | top-1 accuracy|
> | :-: | :-: | :-: | :-: | :-: | :-: |
> | DeiT-Small |  22.05 M | 4.24 G  | 1.04 ms | 5251MB | 80.69 |
> | ResNet-101 |  44.55 M | 7.85 G  | 1.68 ms | 9682MB | 77.97|
> | Swin-Small |  49.61 M | 8.52 G  | 2.41 ms | 13976 MB | 83.02 |
> | ClusterFormer-Small | 48.71 M | 8.24 G  | 2.24 ms | 13215 MB | 83.41 |
>
> #### **Q2. Initialization of the cluster center**
>
> **A2:** Thank you for your insightful question regarding the potential for further optimization in the initialization of the cluster centers. In our study, we employ the Forgy method for initialization, which involves randomly choosing K data samples as the initial centers. This approach has been selected because of its simplicity and the practicality it provides, allowing us to handle large datasets and complex structures efficiently. While it is random and thus prone to variability, it does, in many cases, provide a reasonable starting point for our EM clustering. Nonetheless, we recognize that this may not always guarantee the optimal solution due to the stochastic nature of the algorithm. As such, we are considering investigating the use of more advanced methods for the initialization of the cluster centers in our future work.
>
> #### **Q3. Value of K**
>
> **A3:** Sorry for the confusion. In our study, we set the value of k to 100 in the context of image classification. The selection of k is indeed related to the specific task. In a broader sense, the choice of k in our model is a balance between model complexity, performance, and computational efficiency. While it is generally true that increasing the value of k can potentially improve model performance by allowing it to capture more intricate patterns within the data, this comes with a trade-off. A larger k leads to a higher number of parameters, which in turn increases the computational demand and potentially the risk of overfitting. Therefore, the selection of k should be guided by the specific requirements and constraints of the task. For different k in different layers, we utilize a progressive pipeline that passes the centers directly between different layers. It may have a different performance if we change the number of k in different layers.
>
>
> #### **Q4. Better performance with the same inference speed**
>
> **A4:** This is a great question. As shown in the above table, our model achieves on-par inference speed compared with the Swin transformer while achieving better performance. A larger model might have higher inference latency.

---

> > ### Comment · Reviewer_x7UH · 2023-08-19
> > **Additional Questions**
> >
> > Thanks for your response. My confusion was partially solved after seeing the table.
> >
> > I appreciate the idea that clustering with EM-like optimization, and that's why I gave a high score.
> > However,
> > 1) I still think the initialization of k needs to be studied.
> > 2) I have the same question with Reviewer bUzy14 why ClusterFormer has a similar number of parameters and FLOPs compared to the Swin transformer?
> > The ClusterFormer uses recursion in the network, which may lower the number of parameters and increase FLOPs.
> > The answer to Reviewer bUzy14  is still confusing to me. Please explain it in more detail.

---

> > > ### Author Response · Authors · 2023-08-19
> > > **Response to the additional questions**
> > >
> > > Thank you for the additional questions! We answer them as follows:
> > >
> > > #### **Q1: Study of k**
> > >
> > > **A1:** Thank you for your valuable suggestion. We fully agree with the reviewer on the importance of exploring the impact of different values of k. In fact, we have conducted experiments in this regard, and selected 'k = 100' as it achieved a favorable balance between performance and efficiency. Below, we provide the experimental results for different 'k' values.
> > >
> > > | Value of K | Parameters | FLOPs  | inference latency | GPU memory | top-1 accuracy|
> > > | :-: | :-: | :-: | :-: | :-: | :-: |
> > > | K = 144 | 30.46 M | 5.60 G  | 1.65 ms | 8.51 G | 81.33 |
> > > | K = 100 | 27.85 M | 4.19 G  | 1.31 ms | 7.79 G | 81.31 |
> > > | K = 49 |  23.13 M | 2.47 G  | 0.87 ms | 7.17 G | 80.93 |
> > > | K = 25 |  20.25 M | 1.35 G  | 0.52 ms | 6.79 G | 79.59 |
> > >
> > > These results illustrate that while a larger 'k' typically yields improved performance, it also comes with a higher computational burden. When 'k' increases to 144, the model's performance plateaus, but at the expense of significantly increased computational resources. This key observation is the primary rationale behind our choice of 'k = 100' for presenting our main results.
> > >
> > > In addition, we conduct an additional experiment of choosing different k in different layers as suggested, i.e., (100, 64, 36, 25) and (25, 36, 64, 100), and compare with (100, 100, 100, 100). The results are shown below.
> > >
> > > | K in different layers | top-1 accuracy|
> > > | :-: | :-: |
> > > | (100, 100, 100, 100)  | 81.31 |
> > > | (100, 64, 36, 25)  | 80.52 |
> > > | (25, 36, 64, 100)  | 80.26 |
> > >
> > > The results suggest that maintaining the same 'k' value across different layers yields the highest level of performance. As we explained earlier, this occurs because our model efficiently transfers the centers directly between consecutive layers. Consequently, employing different numbers of centers in different layers could potentially introduce additional optimization challenges, such as the need to learn an optimal projection network to transition from the 100 centers to the desired 64 centers.
> > >
> > > We will incorporate these supplementary results and discussion in the revised version. Once again, we sincerely appreciate your constructive suggestion!
> > >
> > > #### **Q2: Parameters and FLOPs**
> > >
> > > **A2:** Thank you for your question. We try to provide more comprehensive details here.
> > >
> > > First, our ClusterFormer configuration closely aligns with that of the Swin Transformer, such as having an identical number of blocks in each stage and similar network size, dimensions, and depth. Consequently, the difference in the number of parameters between ClusterFormer and Swin Transformer is relatively small.
> > >
> > > Second, a notable divergence emerges when we consider FLOPs due to the cross-attention clustering mechanism, which differs from the self-attention mechanism in Swin Transformer. Thus, the training FLOPs of ClusterFormer **in a single iteration** is significantly smaller compared with Swin Transformer. The recursion does not increase the number of parameters but introduces extra FLOPs since the parameters are updated in each iteration. As a result, the cumulative FLOP count for ClusterFormer increases and eventually reaches a level close to that of the Swin Transformer, within three iterations or recursions.
> > >
> > > We hope this explanation clarifies your question. Thank you for your valuable feedback!

---

### Official Review · Reviewer_R2Rz · 2023-07-08

**Soundness:** 3 good
**Presentation:** 3 good
**Contribution:** 3 good
**Rating:** 6
**Confidence:** 4

**Summary:**

The paper proposes a ClusterFormer approach for visual recognition. The ClusterFormer has a Recurrent Cross-Attention Clustering stage which aggregates patch-level images features by cross-attention to form so-called "cluster centers" which contains global context information, and a Feature Dispatching which adds global context information from cluster centers to local patch-level features. Results show that the proposed approach obtains better results than the previous approaches on multiple visual recognition tasks, including image classification, object detection, instance segmentation, semantic segmentation, and panoptic segmentation.

**Strengths:**

+ The proposed approach is interesting.

+ Most parts of this paper are easy to understand.

+ Promising results are obtained by the proposed approach.

**Weaknesses:**

- Over-claimed.
The paper claims a "universal vision model". However, adaptations are needed to make the proposed approach work on different tasks. And different fine-tuned models are needed for different tasks. This is not new and exciting for a visual backbone - It's common in visual backbone work that after some adaptation an ImageNet pre-trained model can be adapted to different tasks using different fine-tuned models, from CNN-based models (e.g., ResNet) to transformer-based models (e.g., Swin). Therefore, saying the proposed model as a "universal vision model" is over-claimed. The author(s) should remove this claim in their next version of paper.

- Qualitative results of cluster centers.
It would be better to show some qualitative results of cluster centers to show that the learnt cluster centers are really "cluster centers" rather than just some feature vectors with global context information.

- Others.
a) Should give definitions of abbreviations, e.g., RCA in Eq. (4).
b) The term "recurrent" is not accurate. It's more like a recursive process instead of recurrent.
c) The Recursive process may take a lot of time and computation. The author(s) should provide FLOPS, inference latency, and training/testing GPU memory comparisons with other approaches as well instead of just # parameters.

**Questions:**

What's the FLOPS, inference latency, and training/testing GPU memory costs of the proposed approach compared to other approaches?

**Limitations:**

The authors have adequately addressed the limitations in their supplementary material.

---

> ### Author Rebuttal · Authors · 2023-08-09
>
> #### **Q1. claims a "universal vision model"**
>
> **A1:** We appreciate your invaluable feedback. As you correctly point out, each task requires specific adaptations (a small head) and fine-tuning to maximize performance, which is a common practice in the field. Our intent behind the terminology was to highlight the unique aspect of our framework. Our proposed approach utilizes a straightforward clustering paradigm and this paradigm allows the model to simultaneously tackle heterogeneous tasks, which demonstrates a generic learning capacity --- we, therefore, called it a "universal visual learner".
>
> We agree with your comments that using this term might lead to misunderstandings and may appear as an overclaim. We will revise our terminology to more precisely reflect the model's capability and will stress the necessity of task-specific adaptations in further versions.
>
> Again, we greatly value your feedback as it helps improve the clarity and accuracy of our work.
>
>
> #### **Q2. Qualitative results of cluster centers**
>
> **A2:** We appreciate your careful review and the suggestion to provide qualitative results of cluster centers. In response to your feedback, we would like to draw your attention to Figure 3 in our paper. This figure provides a visualization of the center-feature assignment at the final stage of our recurrent cross-attention clustering process. Each color in the map corresponds to a distinct cluster, representing a grouping of features with similar representations. These visualizations are meant to demonstrate that our cluster centers are representative and serve as meaningful points of aggregation for related feature vectors. The cluster centers act as representative vectors around which similar features coalesce, which is a defining characteristic of "cluster centers" in many clustering algorithms.
>
> #### **Q3. Definition of abbreviations**
>
> **A3:** Sorry for the confusion. RCA stands for the recurrent cross-attention layer. In light of this, we will revise our paper to include more definitions of abbreviations.
>
> #### **Q4. 'Recurrent' is not accurate**
>
> **A4:** Thank you for your insightful feedback. We will use “recursive process” in the version.
>
> #### **Q5. Computation cost**
>
> **A5:** The computation cost and inference speed are reported as follows.
>
> |  Method  | Parameters | FLOPs  | inference latency | GPU memory | top-1 accuracy|
> | :-: | :-: | :-: | :-: | :-: | :-: |
> | DeiT-Tiny |  5.72 M | 1.26 G  | 0.35 ms | 1884 MB | 74.50 |
> | ResNet-50 |  25.56 M | 4.12 G  | 0.96 ms | 7658MB | 76.55 |
> | Swin-Tiny |  28.29 M | 4.36 G  | 1.35 ms | 7990 MB | 81.18 |
> | ClusterFormer-Tiny | 27.85 M | 4.19 G  | 1.31 ms | 7786 MB | 81.31 |
>
> |  Method  | Parameters | FLOPs  | inference latency | GPU memory | top-1 accuracy|
> | :-: | :-: | :-: | :-: | :-: | :-: |
> | DeiT-Small |  22.05 M | 4.24 G  | 1.04 ms | 5251MB | 80.69 |
> | ResNet-101 |  44.55 M | 7.85 G  | 1.68 ms | 9682MB | 77.97|
> | Swin-Small |  49.61 M | 8.52 G  | 2.41 ms | 13976 MB | 83.02 |
> | ClusterFormer-Small | 48.71 M | 8.24 G  | 2.24 ms | 13215 MB | 83.41 |

---

### Official Review · Reviewer_RD2x · 2023-07-08

**Soundness:** 2 fair
**Presentation:** 2 fair
**Contribution:** 2 fair
**Rating:** 6
**Confidence:** 3

**Summary:**

In this paper, ClusterFormer, which is a network for various vision tasks, is proposed. The proposed algorithm consists of two major components: recurrent cross-attention clustering and feature dispatching. The recurrent cross-attention clustering module groups similar patches by combining EM algorithm with the attention technique. Also, feature dispatching module updates the feature based on clustering results. These two modules are alternately employed. To evaluate the performances of the proposed algorithm, results on various vision tasks, including image classification, object detection, and some segmentations, are provided. In most tests, the proposed algorithm outperforms the conventional techniques.


**Strengths:**

1. The proposed algorithm is simple, but technically sound. The design of each core module may not be very new to the vision community. However, I think it is still meaningful when considering the experimental results.

2. It seems like reproducible since the authors provide the demo code as well.

3. Extensive experimental results are provided. The proposed algorithm achieves the best scores in most tests. Also, the detailed ablation studies are included.

**Weaknesses:**

1. In section 4, more detailed discussion on the experimetal results is needed. In 4.1.1-4.1.5, only the performance gains, which may be easily recognized by readers, are described continuously. It would be useful if the discussion about the major difference between the proposed algorithm and the conventional algorithms, which causes the performance gaps, is provided.

2. The proposed algorithm omits the explanation on some parts. For example. in Eq. (5) on page 4, there is no definition for FFN. Then, L288  on page 7, it is described. Even though FFN is widely-used these days, it would be better to explain briefly at least. Similarly, there is no explanation for adaptive sampling or adaptive pooling in L158.



**Questions:**

I think that the proposed algorithm has more strengths than weaknesses in overall, even though the each piece of the proposed algorithm may seems not very new to the vision community. It is mainly because the proposed algorithm achieves the improved scores in various tests. Also, it is because the proposed algorithm is simple in some way, but sound to me. However, I'm still willing to see other reviewers' comments and the author responses.

**Limitations:**

The authors have addressed the limitation and broader impacts in the Appendix.

---

> ### Author Rebuttal · Authors · 2023-08-09
>
> #### **Q1. Detailed discussion**
>
> **A1:** We thank the reviewer for the feedback. The performance gains we observed are predominantly due to our core design of recurrent cross-attention and feature dispatching mechanisms. This strategy enables an implicit generation of semantically rich cluster centers and subsequently distributes this information individually to each output token. In specific, this cross-attention clustering mechanism allows our model to identify cluster centers and utilize feature dispatching to update the corresponding cluster center's feature representations. This innovative process inherently elevates our model's performance compared to traditional methodologies, as observed in our experimental results.
>
> To better address your concern, we will ensure that these key differences and their impact on the performance gaps are discussed in detail in our revised manuscript. Again, we greatly appreciate your constructive feedback.
>
>
> #### **Q2. Explanation of FFN and adaptive pooling**
>
> **A2:** To clarify, FFN stands for Position-wise Feedforward Network which is an integral part of the Transformer architecture. It comprises two fully connected layers along with an activation function used in the hidden layer.
>
> In addition, adaptive pooling is a type of pooling that adjusts to the size of the input feature map. This differs from max pooling and average pooling, which require a fixed size of the pooling window. Adaptive pooling calculates an appropriate window size to achieve a desired output size, offering more flexibility and precision compared to traditional pooling methods.
>
> We will ensure that a proper definition of these terminologies is provided in the revised manuscript.

---

> > ### Comment · Reviewer_RD2x · 2023-08-16
> >
> > Thank you for your response. As other reviewers pointed out, the novelty of the proposed algorithm may be limited according to the point of view. However, I'm still think that this paper is interesting to the community, because the proposed algorithm achieves the high scores on various major benchmarks -- even though the proposed algorithm consists of familiar modules and concepts. Also, the authors provide the demo code and it will be helpful for reproducing. Therefore, I decide to keep my original rating.

---

> > > ### Author Response · Authors · 2023-08-17
> > > **Thanks for your valuable feedback and support**
> > >
> > > Dear Reviewer,
> > >
> > > We deeply appreciate your insightful feedback and your acknowledgment of the contribution of our paper. We will make sure to incorporate all the suggestions in the revised manuscript. Your support and recognition are valuable to us.
> > >
> > > Thank you once again.
> > >
> > > Best,
> > >
> > > Authors

---

### Official Review · Reviewer_bUzy · 2023-07-21

**Soundness:** 2 fair
**Presentation:** 1 poor
**Contribution:** 3 good
**Rating:** 5
**Confidence:** 5

**Summary:**

The paper proposes a new architecture ClusterFormer.  Instead of a transformer block, ClusterFormer uses recursive clustering implemented by cross-attention layers and MLP layers dispatching cluster information to tokens. ClusterFormer outperforms other architectures in the same number of parameters.


**Strengths:**

- ClusterFormer uses cross-attention similar to k-means clustering, significantly different from a self-attention layer of ViT. It enhances the originality of research and makes ClusterFormer interesting.

**Weaknesses:**

- Computation (FLOPs) and throughput of ClusterFormer are not reported. Comparison in computation costs is necessary for architecture paper.
- The paper lacks essential details regarding network architectures, such as network depth, channels, stage configuration, and number of iterations for clustering. I don't think the paper includes enough information to reproduce the results.
- Writing should be improved. Because the paper only focused on detailed modules, there is no overall architecture description. It is really hard to figure out ClusterFormer as an architecture.

**Questions:**

- In recent architecture research, computation costs of architecture are the most important part.  How much computation does ClusterFormer require?

- In line 142, the paper claim that ClusterFormer is efficient architecture because $TK << HW$. However, in experiments, $K=100, 150$ are used. Original ViT uses $HW=196$. Although $T$ is not given in the paper, I think it is not enough to argue $TK << HW$. Please explain this mismatch between the method and experiments.

- In the first plot of Figure 1, the x-axis scale looks wrong. Improvement of ClusterFormer is just +0.4. But, it looks +4.0 in the plot. Please correct this.

**Limitations:**

.

---

> ### Author Rebuttal · Authors · 2023-08-09
>
> #### **Q1. Computation Cost**
>
> **A1:** Thank you for the suggestion. The computation cost and inference speed are reported as follows. We will incorporate them in the appendix.
>
> |  Method  | Parameters | FLOPs  | inference latency | GPU memory | top-1 accuracy|
> | :-: | :-: | :-: | :-: | :-: | :-: |
> | DeiT-Tiny |  5.72 M | 1.26 G  | 0.35 ms | 1884 MB | 74.50 |
> | ResNet-50 |  25.56 M | 4.12 G  | 0.96 ms | 7658MB | 76.55 |
> | Swin-Tiny |  28.29 M | 4.36 G  | 1.35 ms | 7990 MB | 81.18 |
> | ClusterFormer-Tiny | 27.85 M | 4.19 G  | 1.31 ms | 7786 MB | 81.31 |
>
> |  Method  | Parameters | FLOPs  | inference latency | GPU memory | top-1 accuracy|
> | :-: | :-: | :-: | :-: | :-: | :-: |
> | DeiT-Small |  22.05 M | 4.24 G  | 1.04 ms | 5251MB | 80.69 |
> | ResNet-101 |  44.55 M | 7.85 G  | 1.68 ms | 9682MB | 77.97|
> | Swin-Small |  49.61 M | 8.52 G  | 2.41 ms | 13976 MB | 83.02 |
> | ClusterFormer-Small | 48.71 M | 8.24 G  | 2.24 ms | 13215 MB | 83.41 |
>
> #### **Q2. Lack essential details**
>
> **A2:** Thank you for the feedback. In Section 3.3 of the paper, we try to comprehensively outline the implementation details. Furthermore, Fig 2 of the paper provides a visual depiction of the overall network architecture, illustrating its structure in a more intuitive way.
>
> The channel configuration, which you pointed out as a missing detail, is described in our ablation study. The study, presented in Table 6, explores the head dimension, and its impact on the results of our experiments. We also studied the number of iterations required for the clustering process and provided these details in the same table.
>
> In an effort to provide additional clarity and aid in reproducing our results, we have included both pseudo-code and the actual code with anonymous links in the supplemental material.
>
>
> #### **Q3. TK<HW**
>
> **A3:** Thank you for the feedback and we acknowledge your concern about the perceived mismatch between the theoretical claim TK < HW. Instead of using ViT, we follow the most recent pyramid architecture (e.g., Swin Transformer or Pyramid Vision Transformer) for building our model. Considering the nature of the pyramid architecture during the encoding process, the effective HW varies across different stages, with values of 12544, 3136, 784, and 196. The value of HW=196 applies to the final stage, while for earlier stages, HW can be significantly larger. Considering this, the efficiency of our model should be understood in the context of these pyramid architectures, where TK can indeed be much smaller than HW, especially in the earlier stages.
>
> #### **Q4. Figure 1**
>
> **A4:** Thank you for the feedback. We will update the figure in the revised version to make it more clear.

---

> > ### Comment · Reviewer_bUzy · 2023-08-14
> >
> > Hi, thank you for your response.
> > I read your responses, and here are additional questions.
> >
> > **Q1. Computation cost**
> >
> > The report on computation costs will significantly improve the contribution of your paper.
> > I will adjust my rating after the discussion.
> >
> > I have one more question on the computation cost.
> > As I understand, ClusterFormer uses recursion in the network, which may lower the number of parameters and increase FLOPs.
> > However, ClusterFormer has a similar number of parameters and FLOPs compared to Swin transformer.
> >
> > Can you explain this to me?
> >
> > Does ClusterFormer use more blocks than Swin on stage 4?
> >
> > **Q2. Lack essential details**
> >
> > I missed the number on the ablation study.
> > It would be better to mention the number of heads and recursion at the main experiments of ClusterFormer.
> >
> > Still, I can't find the depth and number of blocks in each stage of ClusterFormer.
> > It is infeasible to reproduce ClusterFormer without knowledge on network depths.
> > I recommend authors to explain the depth of ClusterFormer like Fig. 3 of Swin Transformer paper.
> > (e.g. Blocks x2 on stage1, Block x 2 on stage2, Block x 6 on stage 3, Block x 2 on stage 4)
> >
> >
> > **Q3. TK<<HW**
> >
> > To my knowledge, Swin Transformer uses 224 x 224 images, and HxW on each stage is `56 x 56`, `28 x 28`, `14 x 14`, `7 x 7`.
> > Thus, HW is `3136`, `784`, `196`, `49`.
> >
> > Considers that `T=3` in Table 5 (b), `TK=300 or 450` is larger than HW in stage 3,4, and it is not significantly smaller than TK in stage 2.
> > Thus, I still think the efficiency of ClusterFormer is only applicable to stage 1. Is there an additional explanation for it?

---

> > > ### Author Response · Authors · 2023-08-16
> > > **Response to Reviewer bUzy**
> > >
> > > Thanks for your further feedback! We are glad that our rebuttal addressed some of your concerns. We answer your additional questions as follows:
> > >
> > > #### **Q1: Computation cost**
> > >
> > > **A1:**  This is a great question! The number of parameters of ClusterFormer closely aligns with that of the Swin Transformer, but with significantly lower training FLOPs within a single iteration. The recursive mechanism within ClusterFormer indeed maintains a consistent parameter count in the network while increasing FLOPs, thereby leading to a total FLOP count akin to that of the Swin Transformer. We hope this clarifies your question. We will include the computation cost results along with the elucidation provided above in the revision. We deeply appreciate your insightful suggestion.
> > >
> > > #### **Q2: Regarding the details**
> > >
> > > **A2:** We followed the architecture and configuration of Swin Transformer. For example, for a tiny model, we utilize {2, 2, 6, 2} blocks and {3, 6, 12, 24} heads with head dimensions of 32 as default for different stages, respectively. We will mention these details in the main experiment as suggested.
> > >
> > > #### **Q3: TK<<HW**
> > >
> > > **A3:** Thank you for your feedback. As you correctly pointed out, the average number (over four stages) of HW in the Swin Transformer is 1041, which is approximately 3.5 times larger than TK (300). Moreover, when considering downstream tasks like object detection or segmentation, there's a common tendency to employ higher-resolution images. In such cases, the disparity between HW and TK becomes even more pronounced. Furthermore, we also investigate the tiny model with small numbers of clusters K (better FLOPs with certain performance drop).
> > >
> > > | Method | Parameters | FLOPs  | inference latency | GPU memory | top-1 accuracy|
> > > | :-: | :-: | :-: | :-: | :-: | :-: |
> > > | Swin-Tiny |  28.29 M | 4.36 G  | 1.35 ms | 7990 MB | 81.18 |
> > > | ClusterFormer-K-100 | 27.85 M | 4.19 G  | 1.31 ms | 7786 MB | 81.31 |
> > > | ClusterFormer-K-49 |  23.13 M | 2.97 G  | 0.87 ms | 7172 MB | 80.93 |
> > > | ClusterFormer-K-25 |  20.25 M | 2.35 G  | 0.52 ms | 6793 MB | 79.59 |
> > >
> > > Thanks again for your thoughtful comments! We are happy to discuss more if you have any other questions.

---

> > > > ### Comment · Reviewer_bUzy · 2023-08-16
> > > >
> > > > Thank you for your response.
> > > >
> > > > My two major concerns have been solved.
> > > > 1. FLOPS, latency
> > > > 2. Architecture details
> > > >
> > > > I hope these will be added to the paper correctly.
> > > >
> > > > I think it is a valid improvement in network architecture.
> > > > But, I still believe ClusterFormer doesn't have significant advantages.
> > > > Thus, I adjusted my rating to borderline accept.

---

> > > > > ### Author Response · Authors · 2023-08-17
> > > > > **Thanks for your recognition and positive feedback**
> > > > >
> > > > > Dear Reviewer,
> > > > >
> > > > > Thank you for taking the time to review our response and for the feedback you've provided. We assure you that these points will be incorporated into the paper and deeply appreciate your recognition and positive feedback. This encourages us to push the boundaries further in our research.
> > > > >
> > > > > Once again, thank you for your constructive insights.
> > > > >
> > > > > Best,
> > > > >
> > > > > Authors

---

### Official Review · Reviewer_KGpU · 2023-07-21

**Soundness:** 3 good
**Presentation:** 4 excellent
**Contribution:** 2 fair
**Rating:** 4
**Confidence:** 5

**Summary:**

This paper presents ClusterFormer, a new module as a replacement of the self-attention module for vision transformers. The method involves performing (iterative) clustering between the input tokens and finally summarizes them into a few clusters for feature computation (using a method named feature dispatching). ClusterFormer was validated effective in a wide range of vision tasks including image classification, object detection, semantic/instance/panoptic segmentation, etc.

**Strengths:**

+ The proposed ClusterFormer has been validated on a wide range of vision tasks.

+ The paper is well-written and organized.

**Weaknesses:**

- The proposed ClusterFormer is yet another form of self-attention in which a few clustering tokens were constructed to collect information from visual tokens and then propagate information to them. The novelty is limited given the following works published previously (and it is possible to find more).

[A] Zheng et al., End-to-End Object Detection with Adaptive Clustering Transformer, BMVC 2021.
[B] Fang et al., Msg-transformer: Exchanging local spatial information by manipulating messenger tokens, CVPR 2022.
[C] Liang et al., Expediting Large-Scale Vision Transformer for Dense Prediction without Fine-tuning, NeurIPS 2022.

- The ablative study part mostly studies the design choices (e.g. how to perform feature dispatching), but it misses a study on how the method improves the baseline approaches (e.g. compared against Swin). I am not sure why the proposed method is better than the original vision transformers (the current explanations, including visualization, are insufficient to claim the advantages).

- I think the paper somewhat overclaims the advantages, such as explainability. I am a bit conservative in saying that the proposed method is explainable because the essence is still self-attention.

- I hope to make sure if all methods are compared fairly. For example, in Table 1, the ClusterFormer entries were trained with a batch size of 1024 on 16 A100 cards, what about others (e.g. the closest competitor, Swin)? I know that the batch size can largely impact the final results.

**Questions:**

Please address the concerns raised in the weakness part.

**Limitations:**

Overall, this is an incremental improvement over the original vision transformers. There are two main limitations, including the existence of similar prior methods and the lack of ability to deal with smaller units (e.g. if a token occupies two semantic regions, it is not possible to split the token -- it is also a weakness of prior methods).

---

> ### Author Rebuttal · Authors · 2023-08-09
>
> #### **Q1. ClusterFormer is yet another form of self-attention. The novelty is limited.**
>
> **A1:** We appreciate the reviewer's insightful feedback and we will add discussion with these methods in our version. In the meanwhile, it is important to distinguish our method from the referenced works. Our ClusterFormer adopts the recurrent **cross-attention** mechanism from the perspective of EM clustering to unify the encoding process. Therefore, though our objective of clustering might be similar to [ref1], [ref2], and [ref3], the way in which we accomplish this is conceptually and operationally different. The novelty of our approach lies in the use of this advanced architecture to serve as a universal visual learner.
>
> Specifically, the method in [ref1] uses an adaptive clustering transformer only after the implementation of CNN backbones for further decoding, a different approach than ours that integrates the clustering mechanism throughout the attention mechanism. In [ref2], their methodology involves using multi-head self-attention, a shuffle module, and an MLP to generate messengers in every region. While this approach shares a similarity in the usage of self-attention, it does not align with our novel use of expectation-maximization clustering in our cross-attention mechanism. [ref3] constructs their clustering layer by following the improved SLIC scheme, a distinctly different approach from ours.
>
> Our methodology not only incorporates a different process of implementation but also provides a significant leap forward in the way we leverage the attention mechanism, fundamentally changing the way we gather, process, and disseminate feature representations within our model.
>
> We hope that this explanation provides more insight into the innovative nature of our approach. Thanks.
>
> [ref1] Zheng et al., End-to-End Object Detection with Adaptive Clustering Transformer, BMVC 2021.
>
> [ref2] Fang et al., Msg-transformer: Exchanging local spatial information by manipulating messenger tokens, CVPR 2022.
>
> [ref3] Liang et al., Expediting Large-Scale Vision Transformer for Dense Prediction without Fine-tuning, NeurIPS 2022.
>
> #### **Q2. Difference between original vision transformers.**
>
> **A2:** We appreciate the reviewer's feedback. Our work significantly diverges from the traditional vision transformer architectures such as Swin, ViT, and PVT. These architectures rely heavily on a self-attention mechanism, whereas our ClusterFormer employs recurrent cross-attention clustering. Traditional self-attention mechanisms may struggle with the encoding process which tends to be highly distributed and entangled, making it difficult to disentangle what aspects of the input contribute to the output. In contrast, by cross-attention clustering, our model offers an implicit way to generate the cluster center with high semantics and distribute it to the output token individually. Specifically, this unique process allows us to acquire cluster centers and then use feature dispatching to update the feature representations from corresponding cluster centers.
>
> We believe that our distinct methodological differences inherently lead to improvements over these methods. The recurrent cross-attention clustering offers a more dynamic and flexible means of processing visual knowledge, allowing for more robust and accurate representations.
>
> Thank you for your great suggestion. We will provide more discussion together with an ablation result to better explain and visualize the advantage of our model over traditional vision transformers.
>
> #### **Q3. Explainability.**
>
> **A3:** The explainability root on the nature of the centers for each cluster of feature representations (as shown in Fig 3). The key point is the centrality of the 'centers' for each cluster of feature representations in our model. The main idea is that these cluster centers, which are determined through our recurrent cross-attention clustering process, represent a 'prototype' of the features they cluster. This feature allows us to identify what representation is most salient within each cluster, providing interpretability not typically associated with traditional self-attention mechanisms.
>
> #### **Q4. Fair Comparison.**
>
> **A4:** Thank you for the great question. We used the same batch size and endeavored to maintain a consistent and fair environment for all models to be tested. All experiments are followed by the same training schedulers in mmclassification to ensure a fair comparison. We understand that this point might not have been clearly communicated in our paper. We will make sure to provide more explicit details about the setup of all methods in our experiments during revision.

---

> > ### Comment · Reviewer_KGpU · 2023-08-14
> > **I am still negative on this paper**
> >
> > I read the authors' rebuttal and other reviewers' comments. Overall, I am still negative on this paper.
> >
> > First, I would like to say that I do not totally agree with Reviewer x7UH about the novelty of this work. Using clustering or similar methods in vision transformers is not a new idea. Besides what I meantioned in the review, there are also other methods for improving the speed of vision transformers such as [D][E]. Note that these methods did not use clustering, but also used the relationship between tokens to eliminate less important ones. On the other side, I do not agree with Reviewer bUzy who criticized too much because of the writing issues of this paper.
> >
> > - [D] Rao et al., DynamicViT: Efficient Vision Transformers with Dynamic Token Sparsification, NeurIPS 2021.
> > - [E] Bolya et al., Token Merging: Your ViT But Faster, ICLR 2023.
> >
> > To me, this paper is a borderline case. I shall say that the paper suffers a bit (making me negative) because the authors always tried to overclaim the contributions or results. Many reviewers mentioned about the potential overclaim in the paper. Even in the rebuttal, I am still seeing many words like "a significant leap forward", which somewhat make other statements less convincing.
> >
> > Regarding the rebuttal (to my part) itself, I appreciate the efforts that the reviewers made to address my concerns especially in the relationship to previous methods. After reading it, I am even more confident about my original comments: this is a borderline paper which made marginal contributions. BTW, there are too many "we will"s in the overall rebuttal but few of them were really provided.
> >
> > I choose to keep my original rating.

---

> > > ### Author Response · Authors · 2023-08-16
> > > **Response to Reviewer KGpU**
> > >
> > > Thank you for taking the time to articulate your thoughts on our paper. We truly value your feedback and sincerely seek your approval.
> > >
> > > In terms of our contribution, we value the impartial and forthright evaluation provided by the reviewer. However, we still wish to underscore the significance of our work. While the foundational concept of utilizing clustering within vision transformers might not be new, our innovation lies in refining the recurrent cross-attention clustering with EM-like optimization. This novel approach provides a fresh outlook on feature representation learning, particularly adept at addressing a wide spectrum of visual tasks characterized by varying clustering complexities. Our endeavor represents one of the initial strides toward formulating a universal visual learner, and we hope to inspire future exploration in this direction.
> > >
> > > As for the use of "we will" in our general response, we have tried our best to address the concerns of all reviewers, where all points are incorporated into each individual response to the reviewers. Unfortunately, the guidelines of the rebuttal process have constrained us from directly implementing changes in the manuscript. We commit to fulfilling all the adjustments promised during the revision.
> > >
> > > Thanks again for your thoughtful feedback! We are happy to discuss more if you have any other questions.

---

> > > > ### Author Response · Authors · 2023-08-17
> > > > **Looking forward to further discussion**
> > > >
> > > > Dear Reviewer,
> > > >
> > > > We are pleased to share that we have received approval from the majority of reviewers. We sincerely hope to gain your endorsement as well, since we value a lot of your opinions and feedback. With humility, we respectfully request you to reconsider our endeavor — creating a simple yet elegant universal vision model, an area that has hitherto shown less advancement compared to the realm of language. Our deliverables hold the potential to provide the community with valuable insights for generic vision intelligence. As there remains a brief window before the closure of the discussion period, we would appreciate an open dialogue with you to address any concerns you may have. Your thoughtful engagement is highly valued.
> > > >
> > > > Thank you for your time and consideration.
> > > >
> > > > Best,
> > > >
> > > > Authors

---

### Author Rebuttal · Authors · 2023-08-09

#### **To all reviewers：**

Thank you very much for your valuable time and suggestive comments. We will revise our paper according to your comments. The major changes are as follows:

1. We will offer a more detailed discussion regarding our novelty compared to existing methods and highlight the architecture distinction with original vision transformers as suggested by Reviewer KGpU and QRN3.

2. We will provide more insights on the explainability of our design from the perspective of clustering, as suggested by Reviewer KGpU.

3. We will further clarify the design of the recursive cross-attention mechanism compared with traditional self-attention in any pyramid-based transformer, as suggested by Reviewer bUzy.

4. We will add more descriptions and clarify some misleading terms, as suggested by Reviewer RD2x and R2Rz.

5. We will include more experimental results of the Base-sized model in the revision, according to the suggestion by Reviewer QRN3.

6. We will supplement additional implementation details together with the computational cost to ensure a more complete comparison, according to the comments from the reviewers.

We have strived to address each of your concerns comprehensively and welcome further discussions and insights.

Sincerely yours,

Authors

---

### Author Response · Authors · 2023-08-21
**Summary about author-reviewer discussion**

Dear Area Chair and Reviewers,

We would like to express our sincere gratitude for your efforts in facilitating the discussion regarding our paper. As the discussion is coming to an end, we would like to provide a brief summary of the key points that have been discussed:

1. We have addressed the concerns raised by Reviewer bUzy by providing additional results on computational cost (e.g., FLOPs, latency) as well as offering greater clarity on architecture details. We are pleased to note that Reviewer bUzy has acknowledged that our response adequately addressed their concerns and subsequently increased their score to reflect this.
2. We have provided detailed discussion on experimental results (4.1.1-4.1.5) as suggested by Reviewer RD2x and further clarified FFN and adaptive sampling in the algorithm. We are grateful for Reviewer RD2x’s insightful feedback and the acknowledgment of the contribution of our work.
3. In addition to reporting the number of parameters and computational cost, we have provided additional experimental results on the impact of K in ClusterFormer, addressing the points raised by Reviewer x7UH. We appreciate that Reviewer x7UH maintains their positive assessment.
4. We have incorporated additional experimental results for the Base-sized model according to Reviewer QRN3's advice. We provided further clarification regarding the distinction between SLIC and the recurrent cross-attention mechanism, as well as additional discussion on the explainability of our model. We are grateful that Reviewer QRN3’s concerns are addressed and has subsequently increased their rating.
5. We have clarified certain terms that might have been perceived as misleading, as suggested by Reviewer R2Rz. We have incorporated the suggestion from Reviewer R2Rz to revise some of the statements.
6. We have provided clarification to Reviewer KGpU regarding the distinct novelty and contribution of ClusterFormer, including the architectural differences between ClusterFormer and the original vision transformers. Based on Reviewer KGpU's feedback, we have provided deeper insights into the explainability of our design, especially from the perspective of clustering.

In summary, we would like to express our appreciation to all reviewers for acknowledging our response. We are particularly grateful that Reviewer QRN3 and bUzy have increased their scores, and Reviewer x7UH and RD2x have maintained their positive assessment. We also value the valuable comments provided by Reviewer R2Rz and KGpU. Although we understand that Reviewer R2Rz and KGpU have not engaged in subsequent discussion due to their busy schedule, we believe that our response has effectively addressed their concerns through clear explanations and additional illustrative examples.

We would like to emphasize the contributions of our work, which have been acknowledged by the reviewers and are important to the vision community. While existing language models have largely converged onto a shared paradigm, the realm of vision remains an area ripe for further exploration. Our research represents a concerted effort towards introducing a universal vision model capable of proficiently handling a diverse range of vision-related tasks. We believe that this comprehensive approach offers invaluable insights to the wider community.

We are immensely grateful for the constructive feedback generously provided by the reviewers. In response, we have meticulously refined our work, incorporating the valuable input received. Given the substantial contributions we have made, we anticipate that ClusterFormer can usher in fresh perspectives and make significant strides in advancing the vision community.

Best regards,

The Authors

---

### Decision · Program_Chairs · 2023-09-21

**Decision:**

Accept (poster)

**Comment:**

All but one reviewer agree to accept the paper. The reviewers agreed that while individual components of the proposed approach may not be novel, the overall system is a valuable contribution.

Multiple reviewers raised the concern that the paper made exaggerated claims. The authors are encouraged to carefully delineate the claims, erring on the side of caution. The authors are also encouraged to discuss additional references that have been pointed out by the reviewers.